# Regional adaptation defines sensitivity to future ocean acidification

Piero Calosi[1,2], Sedercor Melatunan[2,3], Lucy M. Turner[2,4], Yuri Artioli[5], Robert L. Davidson[6], Jonathan J. Byrne[6], Mark R. Viant[6,7], Stephen Widdicombe[5] & Simon D. Rundle[2]

Physiological responses to temperature are known to be a major determinant of species distributions and can dictate the sensitivity of populations to global warming. In contrast, little is known about how other major global change drivers, such as ocean acidification (OA), will shape species distributions in the future. Here, by integrating population genetics with experimental data for growth and mineralization, physiology and metabolomics, we demonstrate that the sensitivity of populations of the gastropod *Littorina littorea* to future OA is shaped by regional adaptation. Individuals from populations towards the edges of the natural latitudinal range in the Northeast Atlantic exhibit greater shell dissolution and the inability to upregulate their metabolism when exposed to low pH, thus appearing most sensitive to low seawater pH. Our results suggest that future levels of OA could mediate temperature-driven shifts in species distributions, thereby influencing future biogeography and the functioning of marine ecosystems.

[1] Département de Biologie Chimie et Géographie, Université du Québec à Rimouski, Rimouski, Quebec G5L 3A1, Canada. [2] Marine Biology & Ecology Research Centre, School of Marine Science and Engineering, Plymouth University, Drake Circus, Plymouth, Devon PL4 8AA, UK. [3] Faculty of Fisheries and Marine Science, University of Pattimura, Kampus Poka, Ambon 97233, Indonesia. [4] Department of Marine Sciences, University of Gothenburg, Box 460, Gothenburg 405 30, Sweden. [5] Plymouth Marine Laboratory, Prospect Place, West Hoe, Plymouth PL1 3DH, UK. [6] NERC Biomolecular Analysis Facility-Metabolomics Node (NBAF-B), University of Birmingham, Edgbaston, Birmingham B15 2TT, UK. [7] School of Biosciences, University of Birmingham, Edgbaston, Birmingham B15 2TT, UK. Correspondence and requests for materials should be addressed to P.C. (email: piero_calosi@uqar.ca).

Populations of the same species from different climatic regions often differ in their ability to withstand altered environmental conditions[1,2]. In the marine realm, diversification of populations along thermo-latitudinal gradients has been demonstrated for traits related to thermal physiology in several taxonomic groups[1–6], dispelling the assumption that local adaptation in marine organisms is uncommon. However, there is little evidence as to whether a similar latitudinal trend in sensitivity exists for other environmental drivers, such as elevated partial pressure of $CO_2$ ($P_{CO_2}$)[7].

The anthropogenically driven increase in atmospheric $P_{CO_2}$ is causing a decrease in global ocean pH and altered carbonate chemistry[8]. In several marine invertebrate species, this combination of chemical changes, collectively known as ocean acidification (OA), has been shown to exert significant effects on organismal life history, ecology and behaviour[9,10], and on fundamental physiological processes, resulting in imbalances in acid–base status, a reduction in aerobic scope and a shift in energy budget allocation[11]. These effects can reduce physiological performance, growth and reproductive output[12] and can increase the risk of local extinction, as has been shown to occur in natural high $P_{CO_2}$ habitats[13]. These patterns appear to be partly explained by the ability of some taxa to colonize naturally elevated $P_{CO_2}$ areas by exhibiting greater metabolic control and capacity for acid–base regulation[14–16]. Although there is some evidence for interpopulation variation in the level of sensitivity to $P_{CO_2}$ (ref. 17) and adaptation potential to different $P_{CO_2}$ regimes[18], there has been no large-scale integrative assessment of this variation along latitudinal gradients.

In the current study, we take an integrative approach using data from population genetics, population shell growth and mineralization, physiology and metabolomics to compare the OA responses of different populations of the common intertidal snail *Littorina littorea* (Linnaeus, 1758) sampled from its geographical range in the Northeast Atlantic (that is, from the Iberian Peninsula to the North of Norway[19]; see Fig. 1). Six populations were collected from three distinctive climatic regions: (1) warm temperate; (2) cold temperate; and (3) subpolar (Figs 1 and 2). Our results altogether suggest that different populations of *L. littorea* found across its geographical range of extension were differently sensitive to future OA conditions, with populations closer to the range edges being most sensitive. As a consequence, with increased OA, we may predict a further shift and possible reduction in the geographical range of *L. littorea*, with OA likely modulating in future the species response to ongoing warming. More generally, our study points to the fact that information from single population studies should be used with caution when aiming at predicting species global responses to global drivers. As a consequence, we may be currently over- or underestimating the impact of different environmental changes in different climatic regions, with this having important implications for the development of directives and policies to promote the preservation of marine biodiversity under the ongoing global change.

## Results

**Sequence variability and population structure.** Sequencing of a 598 bp section of COI from 238 individuals revealed 57 haplotypes (Supplementary Table 1). No saturation of substitution was detected in either the entire data set or third codon positions (ISS < ISS.C, $P < 0.001$ in both cases, test of substitution saturation). Percent nucleotide composition was C: 21.9, T: 35.6, A: 22.2 and G: 20.2 and no insertions or deletions were detected. Base frequencies were found to be homologous among sequences at all sites ($\chi^2 = 3.98$, $P = 1.00$, $\chi^2$ test), the 54 variable sites ($\chi^2 = 21.59$, $P = 1.00$, $\chi^2$ test) and the third codon sites ($\chi^2 = 0.13$, $P = 1.00$, $\chi^2$ test).

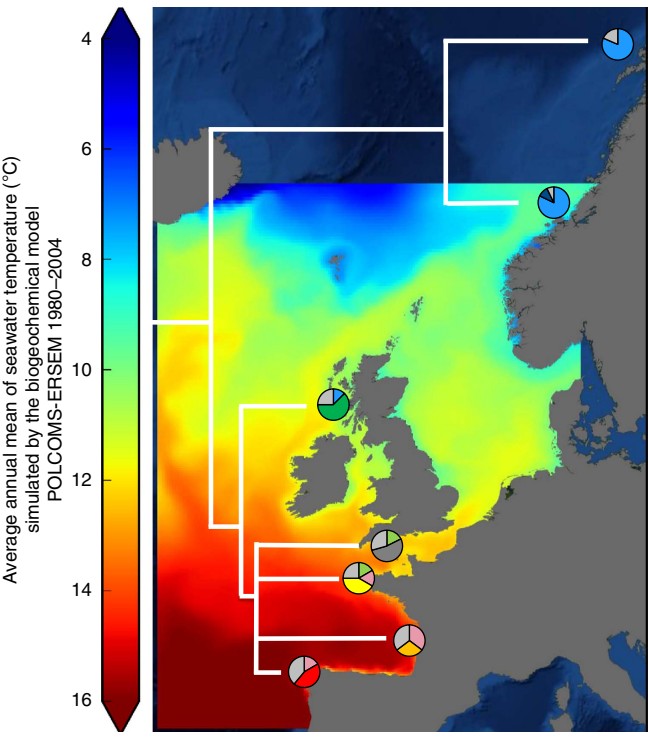

**Figure 1 | Genetic structure of *Littorina littorea* populations.** Thermal map of Europe from the POLCOMS-ERSEM biogeochemical model showing sampling locations and genetic structure of the six different populations of *L. littorea* sampled across the Northwest Atlantic latitudinal gradient (from 42° to 70°): pie chart position indicates location of collection of specimens, from south to north: Vigo (Spain), Île de Ré (France), Roscoff (France), Plymouth (UK), Millport (UK), Trondheim (Norway), Tromsø (Norway). Neighbour-joining population tree. Pie charts = proportion of common haplotypes (grey), shared between climatic regions (pastel colours, northern = blue, mid = green, southern = pink), private to individual sites (primary colours). Background image courtesy of NASA's Earth Observatory.

Overall nucleotide diversity and haplotype diversity were both low and did not differ substantially among sampling sites (Supplementary Table 1). Of the 57 haplotypes detected, two haplotypes (Ll2 and Ll4) were the most common and found at all sites, in most cases constituting over 45% of the total individuals surveyed at each site. However, in the case of the Norwegian sites, Ll1 was the most numerous haplotype present, comprising over 62% of the individuals. In addition, 36 of the 57 haplotypes identified were exclusive (private) to a single sampling site (Supplementary Fig. 1). All of the private haplotypes occurred at low numbers, with only a few individuals present at each sampling site.

The sampled populations had a high level of genetic structuring (Fig. 1 and Supplementary Figs 1 and 2), with a clear genetic difference between the two subpolar and all other sites ($F_{ST}$ values, $P < 0.05$, analysis of molecular variance (AMOVA) test; Supplementary Table 2) and between the climatic regions ($\Phi_{CT} = 0.140$, $P < 0.005$, AMOVA test; Supplementary Table 3).

**Metabolic responses under current $P_{CO_2}$ conditions.** These genetic differences appeared to be related to the physiology of populations. Snails maintained under current $P_{CO_2}$ conditions showed a linear decline in mean metabolic rates (Fig. 3a) and mean metabolomic profiles (Figs 3c, 4 and 5) with increasing latitude. In addition, the presence of a positive linear relationship between mean population metabolomic PC1 scores and metabolic

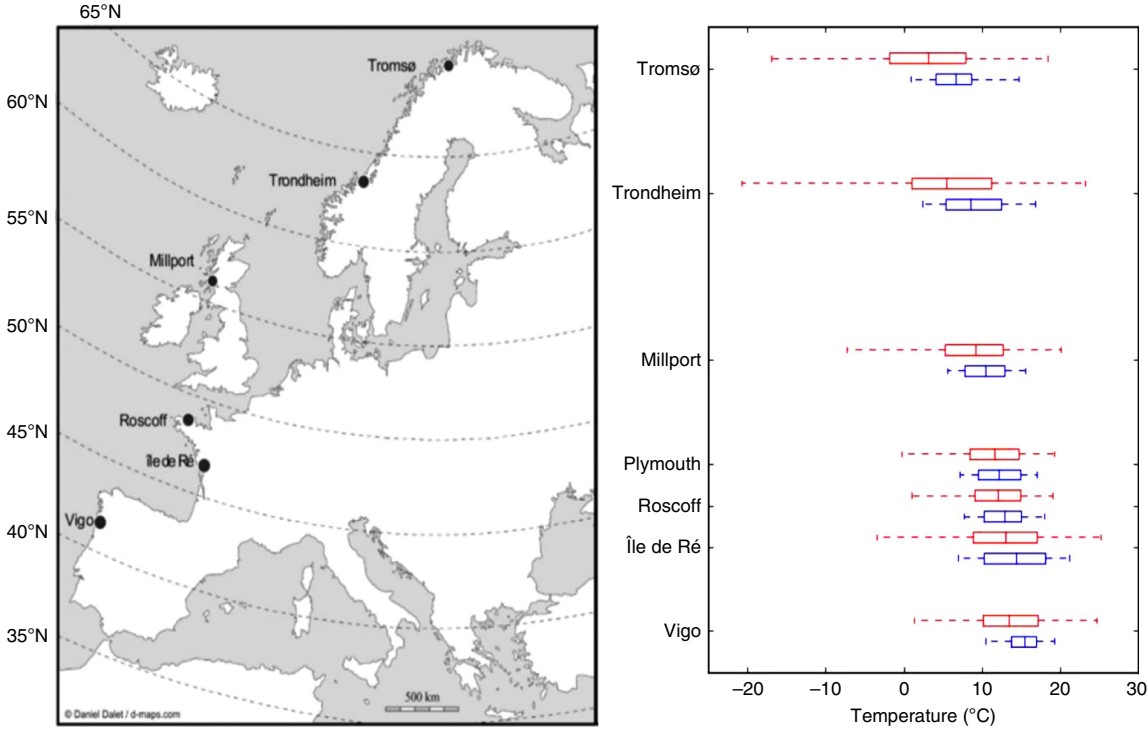

**Figure 2 | Collection locations for populations of *Littorina littorea*.** Map of Europe (left panel) with the locations (full circles) where populations of the common periwinkle *Littorina littorea* (Linnaeus, 1758) were collected from along the Western coastline of Europe during June to August 2010. Horizontal boxes indicate population mean values for temperature (central vertical line) of sea-surface temperature (SST): blue dashed line, and land-surface temperature (LST): red dashed line. The edges define the 25th and 75th percentiles, and error bars represent 95% CI. Temperature data were obtained from the National Oceanic and Atmospheric Administration/National Aeronautics and Space Administration (http://podaac.jpl.nasa.gov/) and comprised the mean annual SSTs for the 5-year period 2005–2009, measured across latitudes from 5.07 to 15.47 °C. Average annual (LSTs were obtained from World Climate (http://www.worldclimate.com/). Map outline courtesy of http://www.d-maps.com/.

rate under control $P_{CO2}$ conditions ($r_5 = 0.084$, $P = 0.019$, best-fit regression approach, see Fig. 3) suggests the existence of a clear mechanistic link between cellular and whole-organism responses (Fig. 3c). This relationship suggests the adaptation of the cellular physiology underpinning the observed whole-organism metabolic rate adaptation along the latitudinal cline investigated (Fig. 3a). The presence of a linear positive relationship between mean population metabolic rate and ATP + ADP levels under control $P_{CO2}$ ($r_5 = 0.77$, $P = 0.041$, best-fit regression approach, $y = 427.820 \times - 1.078$) further corroborate this idea, specifically suggesting the existence of a positive relationship between metabolic rate levels and cellular energy levels[20]. Subpolar (Norwegian) populations had the most divergent metabolomic profiles (Fig. 4a,b) with lower concentrations of homarine and ATP + ADP but higher concentrations of leucine, valine, glutamate, glutamine and formate compared with other sites (for formate and ATP + ADP see Fig. 5b,e). Amino acids are used extensively in energy metabolism[21] and can function as an alternative carbon source in the citric acid cycle when ATP has been exhausted. The ratios of adenylates and amino acids seen here could indicate changes that have occurred in the underlying metabolic machinery of northern populations that have allowed snails to adapt metabolically to cooler climates (Fig. 3a). Furthermore, in marine molluscs, homarine and branched chain amino acids (such as leucine and valine) also underpin processes related to acid–base balance and immune function that are intertwined with energy metabolism. In comparison, glutamate has been highlighted as a possible excitatory neurotransmitter in molluscs[22]. Thus, our data suggest that compared with more southerly populations, subpolar snails significantly upregulate a host of physiological systems,

including metabolism and acid–base balance and immune and neurological function to help maintain homeostasis. In summary, the trends observed under present-day $P_{CO2}$ levels suggest that populations are metabolically adapted to the different environmental conditions found along the latitudinal cline investigated (Figs 1 and 3a,b)[1–3].

**Metabolic responses under elevated $P_{CO2}$ conditions.** Overall, the metabolic rate response under elevated $P_{CO2}$ was nonlinear, with warm-temperate populations (Vigo and Île de Ré) showing a strong downregulation of metabolic rates, cold temperate (Roscoff and Millport) showing upregulation and subpolar (Trondheim and Tromsø) showing no overall differences (Fig. 3a). These results suggest that population levels of metabolic adaptation to prevalent regional environmental conditions lead to different sensitivity to elevated $Pco_2$ conditions, with populations found towards the range edges being more sensitive. Similarly, a greater degree of sensitivity to elevated $Pco_2$ has been proposed for stenotherms compared with eurytherms, based on their different levels of thermal physiological adaptation[23]. In contrast to the pattern observed for metabolic rates, the latitudinal trend in metabolomic profiles under elevated $Pco_2$ was identical to that observed under control $P_{CO2}$ (Fig. 3b). In addition, the presence of a linear positive relationship between mean population metabolic rate and ATP + ADP levels under elevated $P_{CO2}$ ($r_5 = 0.95$, $P = 0.001$, best-fit regression approach, $y = 691.970x - 2.949$), suggests that the positive relationship found under control $P_{CO2}$ conditions between metabolic rate levels and cellular energy levels does not fundamentally change under future OA conditions across the latitudinal range of

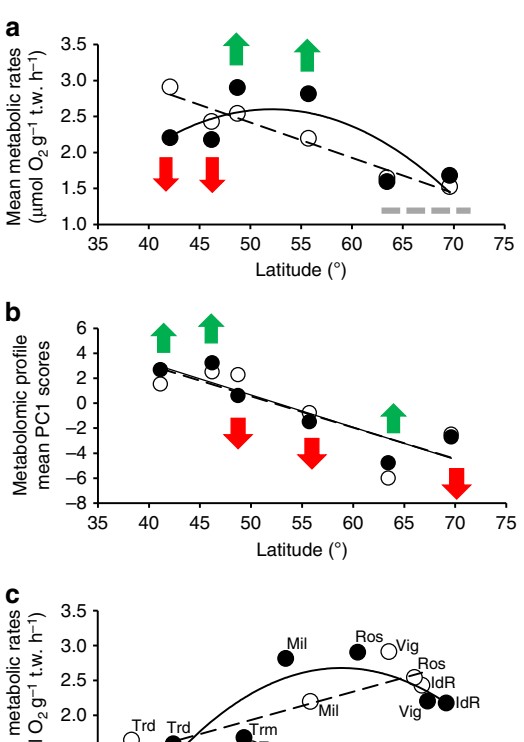

**Figure 3 | Physiological responses of *Littorina littorea* populations.**
(**a**) Mean metabolic rate plotted against the latitude of each population, current $P_{CO2}$, best-fit regression approach $r_s = 0.97$, $P < 0.001$, $y = -0.049x + 4.880$, elevated $P_{CO2}$, $r_s = 0.76$, $P = 0.047$, $y = -0.004x^2 + 0.393x - 7.644$; population collection location latitudinal position provided along the x axis. (**b**) Mean metabolomic profile PC1 scores plotted against the latitude of each population, current $P_{CO2}$, best-fit regression approach $r_s = 0.82$, $P = 0.024$, $y = -0.250x + 13.058$, elevated $P_{CO2}$, $r_s = 0.89$, $P = 0.007$, $y = -0.258x + 13.563$; population collection location latitudinal position provided along the x axis. (**c**) Mean metabolic rate plotted against the metabolomics profile (PC1 score) of each population, current $P_{CO2}$, best-fit regression approach $r_s = 0.84$, $P = 0.019$, $y = -0.135x + 2.272$, elevated $P_{CO2}$, $r_s = 0.89$, $P = 0.007$, $y = -0.053x^2 + 0.014x + 2.677$. Populations: Vig-Vigo, IdR-Île de Ré, Ros-Roscoff, Mil-Millport, Trd-Trondheim, Trm-Tromsø ($n = 6$). White circles and dotted line indicate current $P_{CO2}$ and black circles and full line indicate elevated $P_{CO2}$. Arrows indicate within-population significant increases (green), decreases (red) ($P < 0.05$) or no difference (grey) between different $P_{CO2}$ using *post hoc* Bonferroni test of one-way ANOVA ($P < 0.05$), according to 95% confidence interval test for estimate marginal means (EMM) with Bonferroni correction as reported by Melatunan[59]; t.w. in y-axis labels in **a** and **c** stands for 'tissue wet'.

*L. littorina.* However, a positive binomial relationship was found between mean population metabolomic PC1 scores and metabolic rate under elevated $P_{CO2}$ ($r_s = 0.79$, $P = 0.007$, best-fit regression approach, see Fig. 3c), suggesting that the mechanistic relationship observed under control $P_{CO2}$ conditions between the cellular and whole-organism compartments may change under future OA conditions. Some metabolomic differences for warm-temperate sites were also highlighted by partial least-squares discriminant analysis (PLS-DA) (Fig. 4d–i) under elevated $P_{CO2}$: the Île de Ré population showed a significant decrease in acetate, an essential ingredient for the formation of the co-enzyme acetyl-co-A that is known to be implicated in

anaerobic metabolism of organisms at their thermal limits, and enables energy production[24]; the Vigo population showed a significant decrease in ATP (or ADP) concentrations that is of fundamental importance for central metabolism[25].

**Shell responses under different $P_{CO2}$ conditions.** In order to determine whether differences in metabolic rates and metabolomic profiles following exposure to OA conditions were accompanied by changes in life-history responses[26], growth and (empty) shell dissolution and mineralization status[27] were examined in the six populations investigated (Fig. 6, Supplementary Figs 3–5 and Supplementary Table 4). Under current $P_{CO2}$ conditions there was no relationship between growth rate and latitude (Fig. 6a). In contrast, under elevated $P_{CO2}$, mean growth was significantly negative (between −0.45 and −3.60%) across all populations (Fig. 6a), with this decrease being most pronounced in subpolar and warm-temperate populations (mean reduction −3.25%) compared with cold-temperate populations (−0.46%). Tests of mean passive dissolution on empty shells showed that dissolution and net growth levels were generally comparable in warm- and cold-temperate populations, but were 2.5–4 times greater in subpolar populations (Fig. 6b). It should be noted that differences in the amount of mean shell loss across populations (Fig. 6b) cannot be wholly explained on the basis of differences in mean shell mineral status (that is, original concentration and ratios for shell $Ca^{2+}$, $Mg^{2+}$ and $Sr^{2+}$) nor responses to elevated $P_{CO2}$ (Supplementary Figs 3–5). Exposure to elevated $P_{CO2}$ led to a change in empty shell mineral content of *L. littorea*, although responses varied among populations (see Supplementary Fig. 3), as indicated by the presence of a significant interaction between the factors 'latitudinal position' and '$P_{CO2}$' for $[Ca^{2+}]$ and $[Sr^{2+}]$ (see Supplementary Table 4). In more detail, snails from Roscoff (48.73°) showed the lowest $[Sr^{2+}]$ following exposure to elevated $P_{CO2}$, whereas snails in Trondheim (63.45°) showed the least change. In addition, the populations in Millport (55.73°) and Tromsø (69.62°) showed the highest $[Ca^{2+}]$ after exposure to $P_{CO2}$, whereas snails from Roscoff (48.73°) showed the least change. Shell $[Mg^{2+}]$ and $[K^+]$ were significantly lower in snail exposed to elevated $P_{CO2}$ and both significantly decreased with increasing latitude (Supplementary Fig. 3 and Supplementary Table 4). The ratio $[Mg^{2+}]:[Ca^{2+}]$ and $[Ca^{2+}]:[Sr^{2+}]$ in shells of *L. littorea* decreased and increased significantly following exposure to elevated $P_{CO2}$ respectively, but differently in different population as indicated by the presence of a significant interaction between the factors 'latitudinal position' and '$P_{CO2}$' (see Supplementary Table 4). In addition, the $[Mg^{2+}]:[Sr^{2+}]$ ratio showed a significant decrease with increasing latitude, equally for control and acidified conditions (Supplementary Fig. 4 and Supplementary Table 4). Finally, the presence of a significantly negative relationship between [ATP] and shell $[Mg^{2+}]$, $[K^+]$, $[Mg^{2+}]:[Ca^{2+}]$ and $[Mg^{2+}]:[Sr^{2+}]$ (Supplementary Fig. 5), which appeared comparable at both control and elevated $P_{CO2}$ conditions, could suggest that differences may exist in the way shells are built and in the costs associated with mineralization in each of the different populations of *L. littorea* (Fig. 6).

Nonetheless, the increased shell dissolution rates, as a result of the exposure to sea water with low carbonate saturation states[26], appear to eclipse any potential increase in calcium carbonate deposition resulting in a negative net calcification rate (Fig. 6b)—a mechanism that could explain the negative growth reported under OA conditions. Calculations of population net growth showed that this was 4–8 times greater in subpolar populations compared with the other two climatic regions (Fig. 6c), suggesting that the former upregulate shell mineralization to compensate for

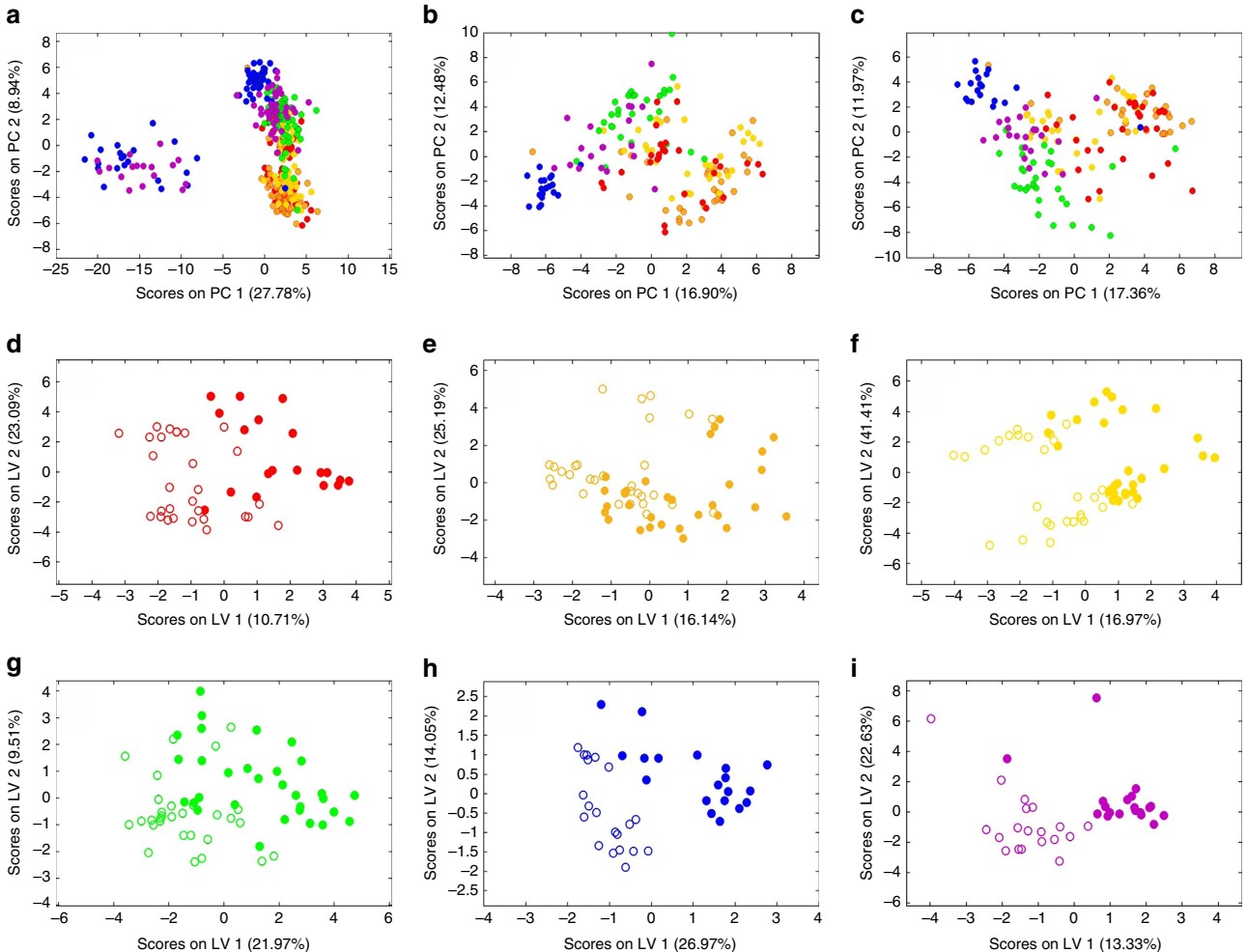

**Figure 4 | Metabolomic profiling of *Littorina littorea* populations.** PCA score plots for (**a**) all populations (both *P*co$_2$) (Vigo-red, Île de Ré-orange, Roscoff-yellow, Millport-green, Trondheim-blue, Tromsø-purple). (**b**) Control *P*co$_2$. (**c**) Elevated *P*co$_2$. (**d**) Vigo, (**e**) Île de Ré, (**f**) Roscoff, (**g**) Millport, (**h**)Trondheim, (**i**)Tromsø PLS-DA score plots for forward selected metabolomic profiles for each population (individual data, $n = 30$). Open circles indicate current *P*co$_2$ and filled circles indicate elevated *P*co$_2$.

extreme levels of shell dissolution as a response to exposure to corrosive waters[28] and, likely, with energetic and metabolic implications[20,26,29].

**Adaptive significance of physiological and life-history responses.** The adaptive significance of the physiological and life-history responses of *L. littorina* populations is supported by the observation that ~50% of each population comprises unique haplotypes (Fig. 1, Supplementary Figs 1 and 2 and Supplementary Tables 1–3)[18]. Adaptation to the warm and cold extremes of a species range appears to cause distinctive stress responses to future OA, confirming the idea that different levels of sensitivity of marine organisms to environmental change is likely based on extant differences in thermal physiological adaptation[23,30]. Under high *P*co$_2$, warm-temperate populations decreased metabolic rates (Fig. 3b) and had a reduced capacity for growth (Fig. 6a) that appear to be due to a reduction in energy metabolism, but could also be caused by an increase in maintenance costs[20,26] and/or by homeostatic tradeoffs between energy metabolism and other fundamental functions. Metabolic depression has evolved as a strategy to minimize the negative effects of acute and chronic environmental extremes, including extreme and variable levels of pH (refs 31 and 32). In contrast, subpolar populations maintained their metabolic rates unchanged under OA conditions (Fig. 3b),

but increased their mineralization effort, likely to compensate for a greater than threefold level of shell passive dissolution than that detected in other populations (Fig. 6c). These processes appear to be associated with altered metabolic functions, as suggested by the distinctiveness of the metabolomes of the sub-polar populations' (Figs 4 and 5), and it is reasonable to believe it is accompanied by a reallocation of the snails' energy budget largely towards growth and mineralization. Functional analyses of transcriptomes of sea urchins collected along a latitudinal pH gradient in the northeast Pacific have confirmed similar molecular patterns for energetically demanding, homeostatic processes in more northerly locations[7]. In contrast, cold-temperate populations showed metabolic rate upregulation when exposed to elevated *P*co$_2$ conditions (Fig. 3b). Such latitudinal differences in metabolic 'strategies' may help explain the observed reduced growth towards range edges (Fig. 6b). Exposure to OA was shown to cause a reduction in the energy metabolism of *L. littorina*[20], and it has been shown that such reductions can lead to a reallocation of the energy budget away from fundamental fitness-related functions, such as growth[29], resulting in smaller adults[16,33]. More in general, our study indicates that a greater physiological ability to deal with environmental extremes may help in retaining higher fitness levels, and thus a species' geographical distribution in the face of environmental changes[2,34], including in the context of OA[14–16,35].

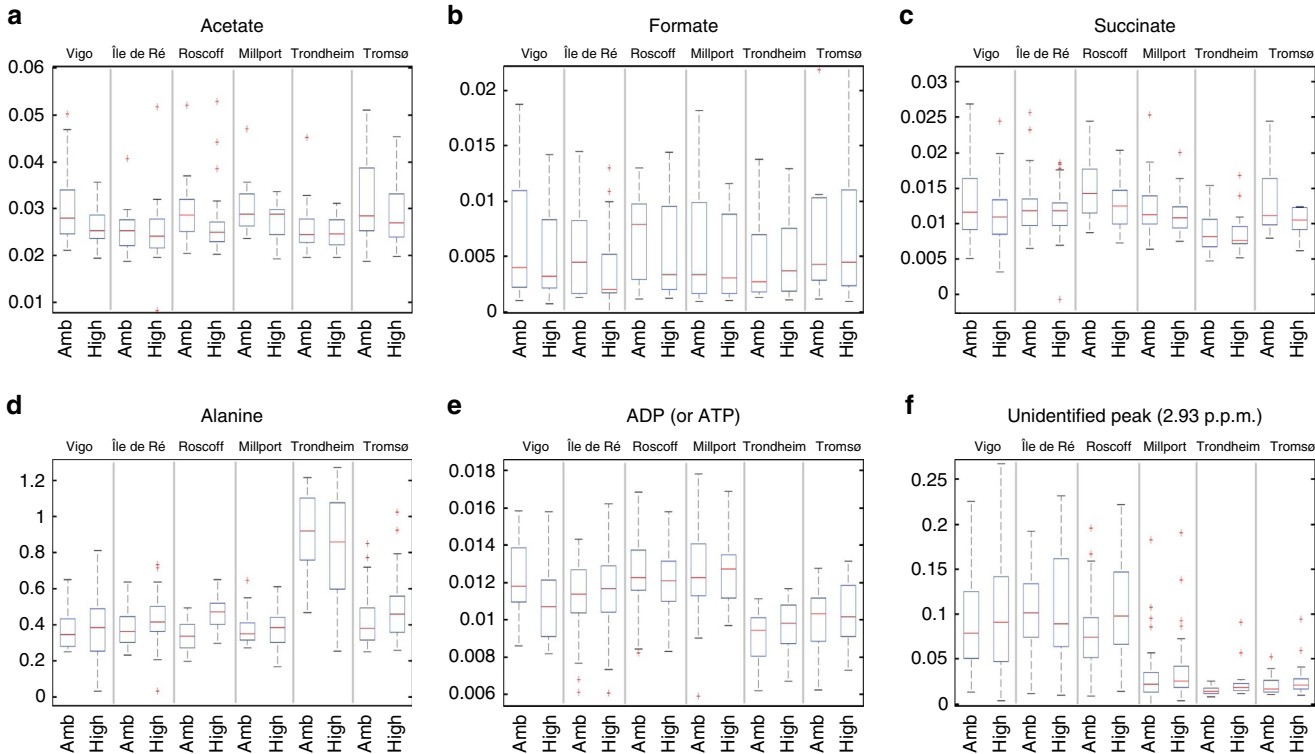

**Figure 5 | Metabolite concentrations of *Littorina littorea* populations. (a)** Acetate, **(b)** formate, **(c)** succinate, **(d)** alanine, **(e)** ADP (or ATP), **(f)** unidentified peak at 2.93 p.p.m. Boxplots describing the integrated peak areas for peaks of interest from each location sampled at different $P_{CO_2}$ conditions. Vertical boxes indicate population mean values for metabolites (central horizontal line), the edges define the 25th and 75th percentiles and error bars represent 95% CI. The symbol '+' represents outlier data. The *y* axis is given in mmol$^{-1}$.

## Discussion

Together, the findings of this study suggest that the relative sensitivity of different populations of *L. littorea* to future OA are likely to vary considerably across its geographical range of extension in the Northeast Atlantic through local and regional adaptation, with populations closer to the range edges being most sensitive. If OA selects against these sensitive, range-edge genotypes, it could cause a reduction of genetic diversity levels that could have far-reaching consequences for the ability of these populations to respond and further adapt to other local and global stressors. As a consequence, with increased OA, we may predict a further shift and possible reduction in the geographical range of *L. littorea*, as genotypes closer to the range edge appear at greater risk of extinction based on their specific metabolic abilities[36]. These genotypes could be replaced by central population genotypes, particularly towards the northern edge where snails appear to be metabolically less competent to deal with OA. Furthermore, the negative relationship between seawater pH and latitude[37] may mean that southern genotypes would require to migrate southward in order to compensate for the negative effects of OA[8], although this seems unlikely given the negative impact of global warming that has already caused a northward migration of *L. littorea* populations at this species' southern range edge[38]. In general, the negative effects of OA on *L. littorea*, a keystone species in rocky intertidal habitats, may have negative repercussions on the structure, dynamics and functioning of intertidal communities and ecosystems across the North Atlantic[39,40], particularly towards the range edges of its distribution.

In conclusion, single population studies should be used with caution when aiming to predict species and community responses to global environmental drivers, such as OA, as phenotypic variation between populations may be considerable. As a consequence, modellers, conservationists, environmental

managers, stakeholders and policy makers should be aware of the pitfalls of developing predictions, directives and policies solely based on single population observations, as has so far been the case[41,42]. Our results show that the effects of environmental changes on species may currently be over- or underestimated in different regions, particularly as, historically, most research is conducted in cold-temperate areas[43], despite polar and tropical species remaining the most vulnerable to climate change[44]. The identification of which populations and genotypes will be more resilient to global change and which ones will be most at risk of extinction will improve conservation efforts and management of natural resources, if marine biodiversity and ecosystem functions are ultimately to be preserved[39,45]. This will be facilitated by the use of an integrated approach, as demonstrated in this study.

## Methods

**Species distribution and specimen collection.** The common periwinkle *L. littorea* (Linnaeus, 1758) is an intertidal gastropod that has a wide latitudinal distribution along the coasts of the North Atlantic, between 42 and 70°N in the eastern North Atlantic (from the Iberian peninsula to the White Sea) and between 38 and 41°N in the Western North Atlantic (from southern New Jersey to northern Labrador)[19,46,47].

A total of six populations living along the European coastline of the north-east Atlantic from three distinctive climatic regions (that is, (1) warm temperate, (2) cold temperate and (3) subpolar) were sampled for this study covering a latitudinal range of 28° (42–70°), from Vigo in northern Spain to Tromsø in northern Norway, a distance of 3,780 km (Figs 1 and 2). During June–September 2010, adults of *L. littorea* were collected by hand from the intertidal zone of rocky shores at low tide. Approximately 250 adult individuals (shell width 12–15 mm) were collected per site and transported to the Marine Biology and Ecology Research Centre at Plymouth University (Plymouth, UK) within 48 h of collection. Before transport, live snails were placed into rectangular plastic containers (20 × 20 × 12 cm) with a perforated lid with 25 holes (ø 10 mm) to allow air circulation and reduce overheating during transportation. Approximately 50–75 individual snails were placed in each container with damp seaweed (*Fucus serratus*) to prevent desiccation and mechanical damage. No mortality was recorded during transport and snails were in good condition with normal activity levels upon arrival

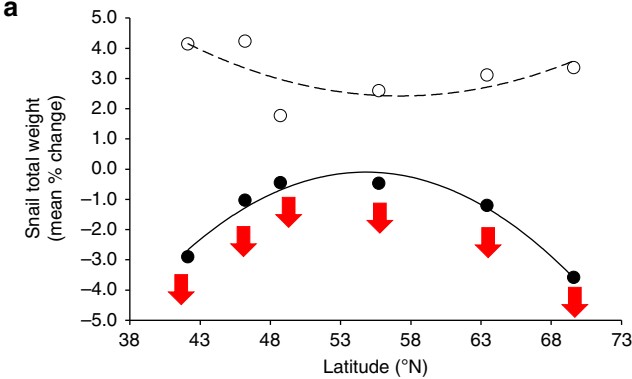

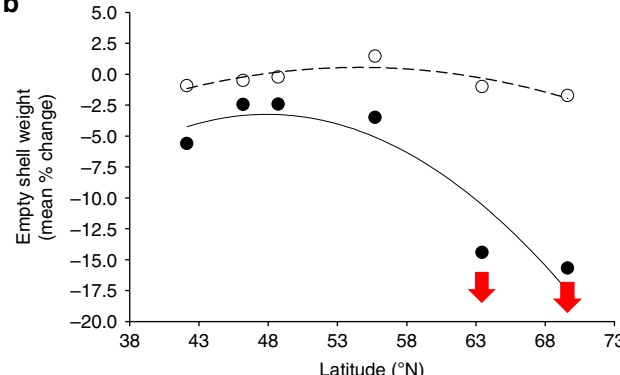

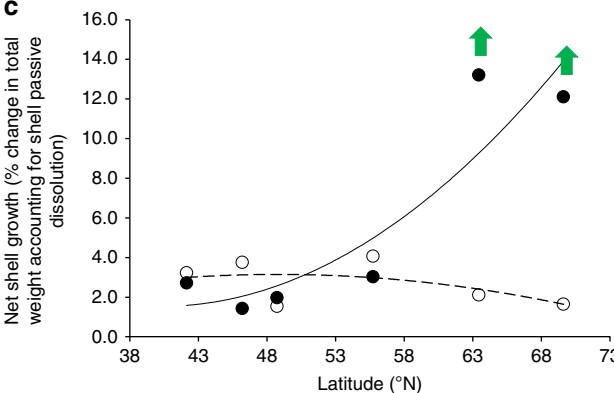

**Figure 6 | Relationship between latitude and growth of *L. littorea* populations.** (**a**) Mean total snail weight, current $P_{CO2}$, best-fit regression approach $r_5 = 0.66$, $P = 0.104$, $y = 0.008x^2 - 0.867x + 27.242$, elevated $P_{CO2}$, $r_5 = 0.98$, $P < 0.001$, $y = -0.016x^2 + 1.753x - 48.143$. (**b**) Mean empty shell weight, current $P_{CO2}$, best-fit regression approach $r_5 = 0.85$, $P = 0.016$, $y = -0.011x^2 + 1.210x - 32.369$, elevated $P_{CO2}$, $r_5 = 0.93$, $P = 0.002$, $y = -0.030x^2 + 2.910x - 72.874$. (**c**) Growth effort (mean total snail weight–mean empty shell weight), current $P_{CO2}$, best-fit regression approach $r_5 = 0.54$, $P = 0.212$, $y = -0.004x^2 + 0.340x - 5.130$, elevated $P_{CO2}$, $r_5 = 0.91$, $P = 0.004$, $y = 0.010x^2 - 1.150x + 24.730$. Population mean data, $n = 6$; population collection location latitudinal position provided along the x axis. White circles and dotted line indicate current $P_{CO2}$ and black circles and full line indicate elevated $P_{CO2}$. Arrows indicate within-population significant increases (green) or decreases (red) ($P < 0.05$, ANOVA test) between current and elevated $P_{CO2}$, and maximum divergence from control conditions observed among all populations using *post hoc* Bonferroni test of one-way ANOVA ($P < 0.05$), according to 95% confidence interval test for estimate marginal means (EMM) with Bonferroni correction as reported by Melatunan[59].

at the laboratory. Once in the laboratory, snails and algae were transferred to aerated filtered sea water (pH 8.01, salinity 35, 15 °C) and maintained for 14 days in four plastic aquaria (56 l, a maximum of 75 individuals per aquarium) before transfer to the experimental setup. Fourteen days of exposure to laboratory conditions was considered sufficient to standardize the snails' physiological condition following any stress from collection and transportation, and to minimize any 'environmental signature' resulting from the local conditions at the collection site. Specimens were fed every second day *ad libitum* on *F. serratus* and 50% seawater volume was replaced every 4 days to maintain stable salinity and pH conditions and to eliminate any accumulated ammonia[20].

All specimens were kept in a temperature-controlled room (12:12 h light/dark regime) at 15 °C. Despite the genetic distinctiveness of the populations of *L. littorina* investigated (Fig. 1), we cannot discount the possibility that the patterns observed in the snails' physiology were generated by exposing northern populations at an experimental temperature (15 °C) that is at the upper end of their natural range of seawater temperatures experienced (particularly for the Tromsø population), but not of air temperatures (Fig. 2). Both these Norwegian populations possess metabolomics fingerprints and metabolic rates (Fig. 3a,b) that are similar to each other, despite snails from these two populations experiencing thermal regimes that are somewhat different (Fig. 2). This suggests that the fairly significant temperature difference between these sites ultimately does not influence their metabolic responses to elevated $P_{CO2}$, adding weight to the idea that the use of 15 °C did not cause bias by stressing the northernmost population.

Because of logistical limitations, population collection was staggered by a week so that no more than two populations were present in the laboratory at the same time. Where possible, collection was organized from south to north in an attempt to collect snails under comparable environmental conditions.

**Experimental design and $CO_2$ manipulation system.** A factorial experimental design incorporating two levels of seawater $P_{CO2}$ representing current conditions (385 µatm, pH 8.0) and predicted OA conditions for the year 2100 (1000 µatm, pH 7.6)[8] was used to investigate the metabolic and metabolomics responses of the snails, and shell dissolution and mineralization to elevated $P_{CO2}$ conditions in the selected populations.

For each population, individual snails were haphazardly allocated to each treatment (120 individuals per $P_{CO2}$ treatment per population) and exposed for 14 days to control laboratory conditions to recover from transport and to adjust to laboratory conditions, before exposure for 14 days to experimental conditions. A $CO_2$/air equilibration microcosm was set up for each treatment in a temperature-controlled room maintained at 15 °C (12 h:12 h, light/dark)[20]. Briefly, each $CO_2$ equilibration system consisted of a header tank (80 l) in which the sea water was either aerated by bubbling normal air or acidified by bubbling pure $CO_2$ gas. From each of the header tanks sea water was continuously gravity fed (600 ml min$^{-1}$) to one of four holding tanks (60 × 35 × 15 cm, 32 l), containing each 30 plastic pots (80 ml). Plastic pots were each provided with 30 holes (4 mm in diameter) to ensure good water circulation, and containing a single snail (shell width 13.67 ± 1.40 mm, mean ± s.d.). The excess water from the four holding tanks flowed into a larger holding tray (simply used to help maintaining temperature stable). From the two holding trays excess water flowed into a sump (a large plastic container size 45 × 36 × 35 cm, 56 l), aerated vigorously to ensure $CO_2$ degasification, and recirculated via a submersible pump (EP68, Hengtong Aquarium Co. Ltd, Hengtong, Taiwan) to both header tanks. $CO_2$ gas was released into the header tank using a multi-stage $CO_2$ regulator (EN ISO 7291, GCE, Worksop, UK) connected to a flip-flop control solenoid valve (ORIFICE 3/16 Closed System, Peter Paul Electronics Co. Inc., New Britain, CT, USA) controlled by a calibrated pH controller (pH-201 Digital, Dream Reef, Humberston, UK). Seawater temperature in the holding tanks was maintained at 15 °C in the temperature-controlled room where the experiment was conducted.

**Aquaria environment monitoring and carbonate system profiles.** Temperature, salinity and $pH_{NBS}$ of sea water in aquaria were measured daily during the experimental period using, respectively, a digital thermometer equipped with a thermocouple (HH806AU, Omega Ltd, Manchester, UK), a handheld refractometer (TA 197 LFMulti350, WTW, Weilheim, Germany) and a pH microelectrode (Seven Easy pH InLab, Mettler-Toledo Ltd, Beaumont Leys, UK) maintained at ambient seawater temperature coupled to a pH meter (Sevengo, Mettler-Toledo Ltd) calibrated using pH standards ($pH_{NBS}$ 4.01, 7.00 and 9.21 at 25 °C, Mettler-Toledo Ltd), also maintained at ambient seawater temperature (Supplementary Table 5). To determine seawater total alkalinity, samples of seawater (volume = 120 ml) were also collected daily and transferred to borosilicate bottles with Teflon caps, poisoned with 30 µl of saturated $HgCl_2$ solution (0.02% sample volume) and kept in the dark until measured by Gran titration (AS-ALK2, Apollo SciTech, Bogart, GA, USA)[20].

Dissolved inorganic carbon, $P_{CO2}$, calcite and aragonite saturation ($\Omega_{calc}$ and $\Omega_{ara}$), bicarbonate and carbonate ion concentration ([$HCO_3^-$] and [$CO_3^{2-}$]) were calculated from pH and total alkalinity measurements using the software program CO2SYS[48] with dissociation constants[49] refitted by[50] and [$KSO_4$] using[51] (Supplementary Table 5).

**Shell growth and empty shell passive dissolution.** In order to determine shell growth under different $P_{CO2}$ conditions the growth of the shell of 18 to 21

individuals per population per treatment was calculated as the proportional change between total weight measurements immediately before and at the end of 14 days of exposure. Snail shells were blotted dry using laboratory tissue paper, and specimens were placed aperture down for 10 min (ref. 52) to allow any remaining water to be expelled via gravity and absorbed with the use of a cotton bud by pressing down gently on the operculum. Each individual was then placed into a plastic vial and weighed using an analytical digital balance (Sartorius 1201 MP2, DWS Inc., Elk Grove, CA, USA)[20]. The percentage change in total weight of intact snails over the experimental exposure was calculated as a proxy for growth.

In order to determine empty shell passive dissolution under different $P_{CO_2}$ conditions, 30 snails were euthanized per population and 15 individuals were assigned to each $P_{CO_2}$ treatment, and shell passive dissolution was calculated as the proportional change between total weight measurements immediately before and at the end of 14 days of exposure. Tissues were immediately carefully removed to avoid damaging the shells that were blotted dry using laboratory tissue paper and placed aperture down for 10 min (ref. 52) to allow any remaining water to be expelled via gravity and absorbed using a cotton bud by pressing down gently on the operculum. Each individual was then placed into a plastic vial and weighed using an analytical digital balance (Sartorius 1201 MP2, DWS Inc.)[20]. The percentage change in total weight of empty shells over the experimental exposure was calculated as a proxy for growth. This measure may represent an overestimate of snail shell passive dissolution but it still represents a reliable standardized estimate for this parameter[27].

**Metabolic rate determination.** Metabolic rates of individual snails were measured at the end of exposure period (14 days) using $O_2$ uptake as a proxy[16]. In order to collect metabolic rate data representative of both the spatiotemporal environmental variability that littorinids are exposed to and their physiological capacity during the phase of maximum stress for their respiratory physiology (that is, low tide), groups of six snails were selected haphazardly from each experimental tank (giving a total of 30 individuals per $P_{CO_2}$ treatment per population combination) and exposed in air in a custom-made incubator (Perspective cube, size $60 \times 60 \times 50$ cm) equipped with an automatic heater (Microclimate Advance Heating System, 500 W, max heat 70 °C, Net Pet Shop Limited, Manchester, UK) that was set to produce one of five temperatures (15, 20, 25, 30 and 35 °C). Exposure lasted for 3 h to mimic an average tide duration. The $CO_2$ in the incubator was adjusted manually according to the level of $P_{CO_2}$ in the treatment conditions either in 385 or 1000 µatm. Up- and downregulation of $P_{CO_2}$ levels was guided through a gas analyser (LI-7000 $CO_2/H_2O$, LI-COR Environmental Ltd, Cambridge, UK) connected to a PC. Following exposure to air each individual snail was placed in a blacked out glass jar (70 ml) for 5 min that was then submersed in a glass aquarium ($49 \times 23 \times 28$ cm, capacity 32 l) containing sea water at 15 °C. Initial $O_2$ uptake was recorded just before the jar was sealed and after 1 h of incubation using a calibrated $O_2$ meter (Model 781, Strathkelvin Instruments, Glasgow, UK) with an $O_2$ electrode (1302 electrode, Strathkelvin Instruments). $O_2$ uptake was expressed as µmol $O_2$ g$^{-1}$ wet weight h$^{-1}$.

**Metabolic profiling.** To assess whether the metabolic profile of snails was influenced by exposure to elevated $P_{CO_2}$, 30 individuals from each of the two $P_{CO_2}$ treatments per population were selected for metabolomics analysis. Immediately following oxygen consumption trials, tissue samples (0.20–0.25 g) of foot muscle were dissected after the shell of each individual was quickly cracked, using a mini grip clamp (Quick-Grip Bar Clamp Q/G5122QC, Hyquip Ltd, Lancashire, UK), and removed. Shells were cleaned in fresh sea water, placed in a labelled plastic vial (Eppendorf tube 1.5 ml, Sigma-Aldrich Co. LLC., Dorset, UK), frozen in liquid nitrogen (temperature of $-196$ °C) and stored in a $-80$ °C freezer.

Metabolites were extracted from tissues and analysed using nuclear magnetic resonance (NMR) spectroscopy-based metabolomics, conducted at the Natural Environment Research Council (NERC) Biomolecular Analysis Facility at the University of Birmingham (NBAF-B). Twelve foot muscles were pooled from sample trays (two foot muscles per tray, in which each tray composed frozen foot tissues from one location), cut into small pieces (60–100 mg) and transferred to pre-cooled Precellys tubes containing steel balls (Precellys tube, cap. 7 ml, Bertin Technologies Corp., Saint Quentin en Yvelines Cedex, France). All procedures were carried out in <10 s. Next, the metabolites were extracted using a two-step solvent extraction protocol as described previously[53]. In brief, MeOH/$H_2O$ (HPLC grade) was added to frozen tissue and homogenized using a Precellys-24 bead-based homogenizer (Stretton Scientific, Stratton, UK; 6,800 r.p.m., $2 \times 20$ s cycles). Homogenized samples were transferred to 1.8 ml glass vials and then CHCl$_3$ (pesticide grade) and $H_2O$ were added. Samples were vortexed for 30 s, cooled in an ice bath for 10 min and then centrifuged (Centrifuge Biofuge Primo-CFH-240–010A, Thermo Scientific Heraeus Corp., Leicestershire, UK; 4,000 g, 10 min, 4 °C). Polar and nonpolar phases were transferred to clean glass vials using a Hamilton syringe and then the polar fractions (400 µl) were dried in a rotary evaporator (SpeedVac SPD111VP115, Thermo Scientific Corp., Surrey, UK) and stored at $-80$ °C until analysis.

All 360 dried polar extracts were analysed using two-dimensional $^1$H, $^1$H J-resolved NMR spectroscopy-based metabolomics (2D pJRES)[54,55]. First, each dried extract was resuspended in sodium phosphate buffer in 90% $H_2O$ and 10% $D_2O$ (0.1 M, pH 7.0) containing 0.5 mmol$^{-1}$ 2,2,3,3-d4-3-(trimethylsilyl)propionic acid (TMSP) (GOSS Scientific Instruments). The 2D J-resolved NMR spectra were

recorded at 500.11 MHz using a Bruker Avance DRX-500 (Bruker, Germany). Spectra were acquired using 8 transients per increment collected into 16k datapoints and a total of 32 increments. Data sets were zero-filled to 128 points in F1. The SEM window function was applied in the direct (chemical shift) dimension and sine-bell in the indirect dimension before Fourier transformation. The chemical shift axis was calibrated to TMSP at 0.0 p.p.m. and the skyline projections (pJRES) calculated[54,55]. All processing was conducted using TopSpin v3.0 (Bruker).

The residual water signal was excluded (4.38–5.22 p.p.m.) and the spectra were binned between 10 and 0.2 p.p.m. (bin width of 0.005 p.p.m.). Next, spectra were normalized using probabilistic quotient normalization and noise filtered (with noise threshold set to 3 times the s.d. of a region of known noise at 9.5–10 p.p.m.). For a bin to be retained, it was required that at least 10 samples have signal above this threshold, resulting in 1,103 bins. Finally, spectra were glog transformed (lambda = 1.38 e$^{-9}$). To assess the technical reproducibility of the analysis, the median relative s.d. for six technical replicates of one biological sample was calculated, equating to 12.7%, comparable with earlier analyses[56]. The relative concentrations of metabolites in the samples were calculated using the web-based automated identification and quantification data mining tool FIMA (Field Independent Metabolite Analysis), developed at the University of Birmingham (http://www.bml-nmr.org/). This software package uses a library of metabolite spectra[55] and outputs a vector of relative metabolite concentrations for each sample.

**Determination of shell mineral content.** The mineral content of empty shells (see above) (for Ca$^{2+}$, Mg$^{2+}$, Sr$^{2+}$, K$^+$) of adult individuals of *L. littorea* exposed for 14 days to different $P_{CO_2}$ conditions was determined using an optical emission spectrometry technique. Shell fragments were meticulously cleaned of all tissue under low-power magnification (10–50, SZXI6 binocular microscope, Olympus, Tokyo) using only plastic or Teflon-coated dissection tools. Shell fragments were then weighed with a high-precision balance (AT201, Mettler-Toledo; degree of precision (dp) 0.01 mg) before being freeze-dried for 24 h (Modulyo freeze drier, Thermo Electron, Waltham, MA, USA) at $-50$ °C. The dry mass of each freeze-dried set of shell fragments was determined with a high-precision balance (AT201, Mettler-Toledo; dp 0.01 mg) before being digested in 2 ml nitric acid (79% concentration, trace analysis grade) in a microwave digestion unit (MarsXpress, CEM, Matthews, NC, USA). Digests were then diluted to 10 ml with ultrapure water and analysed for [Ca$^{2+}$], [Mg$^{2+}$] and [Sr$^{2+}$] with an ICP-Optical Emission Spectrometer (Varian 725-ES, Agilent Technologies, Santa Clara, CA). Shell mineral content is expressed as mmol kg$^{-1}$. Shell mineral content was determined for the populations of Île de Ré, Roscoff, Millport, Trondheim and Tromsø.

**Genetic analyses.** Genetic analyses utilized foot tissue from another 35 specimens of *L. littorea* collected per site in June–September 2010: Tromsø, Norway (Trm) ($n = 32$), Trondheim, Norway (Trd) ($n = 35$), Millport, UK (Mil) ($n = 36$), Plymouth, UK (Ply) ($n = 34$), Roscoff, France (Ros) ($n = 35$), Île de Ré, France (IdR) ($n = 33$) and Vigo, Spain (Vig) ($n = 33$). Whole snails were preserved in 98% ethanol, shipped to the United Kingdom and stored at $-20$ °C before DNA extraction was carried out.

Total genomic DNA was isolated using a modified CTAB protocol[57]. Following DNA extraction, a fragment ($\sim 600$ base pairs) of the COI gene was amplified using two universal primers (that is, LCOI490, 5′-GGTCAACAAATCATAAG ATATTGG-3′ and HCO2198, 5′-TAAACTTCAGGGTGACCAAAAAATCA-3′)[58] using a standard PCR protocol. Sequencing was carried out using the same primers as the initial PCR reactions by Macrogen Inc. (Amsterdam, The Netherlands). Both forward and reverse sequences were generated.

**Statistical analyses.** The relationships between snail metabolic rates, metabolomics profiles and snail growth, empty shell dissolution and mineralization and their latitudinal position along the gradient examined, both under control and elevated $P_{CO_2}$ conditions, were investigated using a best-fit regression approach controlling for both $R^2$ and Akaike information criterion values using a correlative approach[59]. More specifically, only best-fit models are shown. The relationship between latitude and the study variables was investigated under control conditions to shed light on the presence or absence of adaptation to current conditions in the populations investigated, and under future elevated $P_{CO_2}$ conditions to determine the relative sensitivity of the different populations to future $P_{CO_2}$ conditions.

Analyses on shell ion concentrations were conducted using three-way analysis of covariance tests, using the term '$P_{CO_2}$' and 'latitudinal position' as fixed factors, whereas the term 'tank' was set as a random factor that was nested within the $P_{CO_2}$ treatment. In addition, 'snail total weight' or 'tissue weight' were set as covariate. Data always met the assumptions for normality of distribution (in some cases following transformation) (minimum $Z_{176} = 1.290$, $P = 0.072$, Shapiro–Wilk test) and homogeneity of variances (minimum $F_{1,178} = 0.803$, $P = 0.525$, Leven test). The term 'tank' was found to largely have no significant effect on the study variables, but even where it was found to cause a significant effect, removing it from the analyses caused no change to the significance patterns, and thus tank effect was considered marginal and ultimately removed from analyses. Finally, the relationship between the concentration of ATP and shell mineral content (see details on their estimation below) was explored with Pearson's correlation test[26].

Principal components analyses of the processed pJRES NMR data were performed to assess the broad-scale variation between the various groups (three

distinctive climatic regions: (1) warm temperate; (2) cold temperate; and (3) sub-polar, ambient and elevated $P_{CO_2}$ regimes, six field sites) considering all of the metabolites simultaneously. Principal components analyses were conducted using the PLS_Toolbox (version 5.5.1, Eigenvector Research, Manson, WA, USA) within Matlab (version 7.8; The MathsWorks, Natick, MA, USA) following mean centring of the processed NMR data. In order to discover the metabolic responses of the periwinkles to elevated $P_{CO_2}$, supervised multivariate analyses were performed using PLS-DA in the PLS_Toolbox. PLS-DA uses prior knowledge of sample group to maximize separation of the metabolic fingerprints of different groups and derive predictive models. Here, internal cross-validation and permutation testing were used (please see below) to define robust classification error rates associated with the prediction of a metabolic response. Results are presented as 'balanced' error rates, that is, the average of false positive and false negative error rates[60]. Following this, an optimal PLS-DA model was built and a forward-selection approach (applied to the ranked bins, according to the Variable Importance on Projection (VIP)) was used to determine the smallest number of NMR bins that yielded the lowest classification error rate. This forward-selected PLS-DA model was then internally cross-validated (using venetian blinds; repeated 1,000 times) to yield the 'optimal model' classification error rate. Then, in order to evaluate the statistical significance of these error rates, the group labels were randomly permuted and another PLS-DA model was constructed. Internal cross-validation was used to determine a classification error rate based on permutations (repeated 1,000 times). The statistical significance, for the prediction of each group, was assessed by comparing optimal model classification error rate with the null distribution of permuted error rates[61]. Specifically, the number of instances for which the permuted classification error rate was less than the optimal model error rate was calculated and then divided by the total number of permutations (1,000), generating a $P$ value, with $P < 0.05$ indicating that the metabolic fingerprint associated with that group could be discriminated from the other group. Finally, upon selection of the NMR bins of interest, those peaks were identified and then integrated using in-house peak-picking algorithms and the BML-NMR metabolite library[62].

**Phylogenetic analyses.** Sequences were confirmed as *L. littorea* using the Gen-Bank BLASTn search. Sequences were then aligned using ClustalW and edited using BIOEDIT 7.0.5.3 (ref. 63). All alignments and base substitutions were confirmed visually. Homogeneity of base frequencies among sequences was examined using $\chi^2$ tests in PAUP* 4b.10 (ref. 64) and the degree of substitution saturation was examined, using the test of Xia in DAMBE 5.2.15 (ref. 65). The AIC in MODELTEST 3.7 (ref. 66) was used to specify the proportion of invariable sites for this test. Sequence divergences (uncorrected *p*-distance) between haplotypes were calculated using MEGA6 (ref. 67).

Standard population genetic analyses of the sequence data were carried out in ARLEQUIN 3.11 (ref. 68). For each sampling location and for the entire data set the following parameters were calculated: the number of haplotypes ($N_{hap}$), polymorphic sites ($N_{ps}$), average number of nucleotide differences and haplotype ($H_e$) and nucleotide diversity.

Genetic differentiation among sampling locations was investigated using Φ indices in an AMOVA framework in ARLEQUIN[68]. Initially, each sampling site was tested individually for genetic subdivision between samples within populations relative to samples from the whole data set ($\Phi_{ST}$) (tested by permuting haplotypes (10,000) between populations). Two further AMOVAs were calculated to examine genetic subdivision between samples within a group of populations relative to samples from the whole data set ($\Phi_{CT}$) and between samples within populations relative to samples from within the population subgroup ($\Phi_{SC}$). The first of these was on the basis of the three distinctive climatic regions (that is, (1) warm temperate, (2) cold temperate and (3) subpolar) investigated. Individuals from Norway were grouped as 'northern samples', those from the United Kingdom and Roscoff as 'mid-range samples' and those from Île de Ré and Spain as 'southern samples'. The second AMOVA examined northern samples versus mid-range and southern samples combined.

A haplotype network was constructed using TCS 1.21 (ref. 69) to analyse evolutionary relationships among different haplotypes (gene genealogies) (Supplementary Fig. 1). Finally, to analyse genetic differentiation at the population level, a population tree was constructed using the pale lacuna sea snail *Lacuna pallidula* (Lacuninae: Littorinidae) as an outgroup. Genetic differentiation ($D_a$ distances) among populations were calculated using the program DnaSP 5.10.01 (ref. 70) and the population tree was constructed with the neighbour-joining algorithm in PAUP* (ref. 64) (Supplementary Fig. 2).

**Data availability.** COI sequences generated and analysed during this study are available from GenBank www.ncbi.nlm.nih.gov/genbank (accession numbers: KP221321 to KP221558). The data sets generated during and/or analysed during the current study are available on the British Oceanographic Data Centre (BODC) repository at www.bodc.ac.uk (doi:10.5285/40b332e8-e719-40a6-e053-6c86abc012b3).

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

## Acknowledgements

We thank Marie Palmer, Roger Haslam, Ann Torr, Richard Ticehurst, Evelyn Van der Ent and Ulf Sommer for their technical support, and Sophie Martin, Ines Le Fur as well as Maria Elena Alvarez for helping to collect specimens in France and Spain, respectively. We thank Evelyn Van der Ent for her assistance with the collection of genetic data and shell mineralogy data. P.C. acknowledges the great value of many discussions he has had with John Spicer, Dave Bilton and Francisco 'Pancho' Bozinovic over the years on macrophysiology and evolutionary physiology of ectotherms that influenced the development of some of the ideas tested here. We also thank NASA and ECMWF for sharing climatological and oceanographic data. This work is a contribution to the NERC UK. Ocean Acidification Research Programme Benthic Consortium task 1.2 and 1.4. P.C., Y.A. and S.W. were funded by the NERC UK. Ocean Acidification Research Programme (NE/H017127/1). S.M. was funded by the Indonesian Government. L.M.T. was supported by Plymouth University internal funding. P.C., S.M., S.W. and S.D.R. were supported for the metabolomic analyses by the NERC Biomolecular Analysis Facility at the University of Birmingham (R8-H10-61). P.C. is supported by an NSERC Discovery Grant Program and a FRQ-NT New University Researchers Start Up Program Grant. Finally, we acknowledge that our research benefited greatly from the free circulation of people and ideas across the borders of many countries belonging to the European Union.

## Author contributions

S.M., P.C., S.W. and S.D.R. contributed to the rationale for this work and designed the field sampling and experimental work. S.M. carried out the experiments, collected and analysed physiological and growth data supported by P.C., S.W. and S.D.R. M.R.V. led the metabolomic component and S.M., R.L.D. and J.J.B. collected and analysed that dataset. L.M.T. collected and analysed genetic data. P.C. collected and analysed shell mineralogy data. Y.A. produced the biogeochemical model runs used to produce the environmental map in Fig. 1 and Supplementary Fig. 1. P.C., L.M.T. and S.D.R. wrote the first draft of this manuscript, and all authors contributed to the final write up.

## Additional information

**Competing financial interests:** The authors declare no competing financial interests.

