## [Peer Review File · Nature Communications]

Reviewers' comments:

Reviewer #1 (Remarks to the Author):

SUMMARY

This paper presents an integrated set of measurements (respiration, shell mineralogy, dissolution, metabolomics, growth) describing the response of geographically separate populations of the intertidal snail *Littorina littorea* from various locations along the European coastline to pCO₂ at one temperature (15 C). The premise was to define whether these populations show distinct responses to acidification and to explore whether biogeographic boundaries are set by sensitivity to acidification. The authors claim that their results identify population variability, both genetically and physiologically that indicate that range edges will respond differently to acidification, and with different northern and southern population responses. This is a novel set of integrated measurements that tests a hypothesis that, although I believe is accepted by the community, has not been fully articulated or rigorously explored. As such I think the perspective and measurements are of interest to both global change and evolutionary biologists, as well as policy makers and conservationists.

I have one major concern, however, about the design and presentation of the results. All of the experiments were run at 15 C, and it is possible that the response of the northern populations, in particular, is actually an interactive stress response with temperature. As the authors already acknowledge, temperature plays a large part in shaping the distribution of marine ectotherms, and the northern populations, which are shown to have some phylogenetic distinction, may be physiologically unprepared for 15 C. To dispel this concern it would be lovely to have field data providing a range of exposure conditions for the various populations (instead of or augmenting Supplementary figure 1). This would help alleviate the concern that Tromsø snails may never/rarely experience 15 C.

Broadly I agree with the authors that taking population level responses into account is important as we move forward to a more nuanced understanding of the implications of acidification. With regards to that point, I feel that this is a very interesting manuscript, with a lot of novel information, but I am not convinced that there is not an interactive temperature effect that is being hidden by the experimental design. Arguably, however, all similar experiments that were done with temperature establishing the ranges of organisms and testing their sensitivity were similarly flawed in that they did not take into account regional changes in carbonate chemistry, salinity, oxygen, etc that vary along latitudinal gradients. Temperature is clearly one of the (if not THE) strongest driver, which is why it comes up as a red flag here. Redoing the study at local temperatures adds another set of design headaches (how are you sure that you have teased apart temperature and CO₂ if everything is at a different temperature?) and is impractical to ask the authors to do so. I don't actually think that the experimental design is irreparably flawed, I just wish for greater discussion of this point. I also need greater clarification on the methods of the exposures (detailed below) to be able to adequately understand and judge whether the respiration results are attributable to population sensitivity to pCO₂ or to temperature.

GENERAL COMMENTS

I would like to see an expansion of the methods associated with the respiration experiments (from which the shell and metabolomics samples are also derived). The methods of organismal exposure (in air at one of five temperatures between 15-35 C ?? Supplementary File L 127-128) are not well explained, and re-emphasize the previous concern about temperature playing a role in the observed responses of different populations. If all organisms experienced the same temperature regimes (whatever they were, possibly up to 35 C?) they would have been physiologically a larger stressor for individuals from Tromsø (I think - again details on the natural environmental variability here would help reduce the concern).

The second major methodological complaint I have is the use of empty shells to estimate dissolution (reported in L152 of the main text and figure 1c). It has been shown with other mollusks that dead animals dissolve much faster than live ones (Lischka and Riebesell 2012; Manno et al. 2012), which would influence the calculation of mean growth effort. I understand that the authors did not use dead, but rather empty shells, and they may believe that these differences are all due to biological effort on the part of the live organisms, but bacterial growth on empty shells and the presence of protective mucous and protein layers that are present in live individuals may also play a role. The overestimation does not, of course, account for the very large differences seen in dissolution between populations. There is clearly something different (and clearly interesting) about the shells from different regions, I just have concerns that there is no mention of the fact that this may be overestimation.

It is interesting/cool that the ATP/ADP patterns link well with the metabolic rates. There are a number of tantalizing links between the datasets that could be greater highlighted, but due to brevity constraints I recognize that not all of the potential discussion is possible. Still...

Supplementary file: In general I can't figure out your experimental setup. There were 120 critters per treatment and there appears to be one equilibration system per CO₂ treatment that lead into 1 large holding tray per treatment, and then 30 plastic pots with one individual per pot. This means only 30 snails per treatment. Where am I missing something (L67-71)? You either had multiple equilibration systems per treatment or multiple holding trays? This also plays into confused math in line 127 where there is a mention of multiple experimental tanks??? How does that play back into the trays and pots?

Supplementary file section 9.1: what statistical program did you use? When you say that only the best fit model are shown are you referring section 12? This part of the statistics is not very clear to me.

Supplementary file: The three temperature regimes defined in the main text L 78-79 are not defined in the supplementary data before their first mention (L 314) and are then re-defined with different names (L 373-376). Consistency of naming and a reintroduction of the idea before the metabolomics section of the supplementary file would be useful.

Supplementary file: The way that the shell mineralogy results are presented needs some polishing (L 528-541). My confusion has to do with the use of the words lowest/highest and least/most affected as if they were interchangeable. Least effected might be better explained as "showed the least change"? The question is whether you are reporting the absolute values (highest/lowest) or the sensitivity (more or less change). The wording needs cleaned up. The figures also need some clarification (and better captions). I think that the * means that there is an effect of pCO₂ overall, not that there is a bonferoni post-hoc significance of pCO₂ at that particular location, but I could be wrong. The scale of the y axis makes it hard to determine whether there are differences in some of the graphs.

Supplementary file: In generally you need greater consistency in labeling. Sometimes samples are identified by latitude, sometimes by name of site, sometimes by code (country A, B).

Supplementary file: It would be nice if the tables and graphs were always pointed to throughout the text of the supplementary file so that we know which thing is being referenced.

SPECIFIC COMMENTS

Main manuscript

L 81 and L 252-253: the authors are inconsistent with their description of the pCO₂ treatments. Better I think would be to list the achieved levels rather than the desired levels (~420 and 1000 from the supplementary file). Although they did not get as low as they would have liked, 420 is not unreasonable at all for a modern day intertidal organism and is quite reasonable as an "ambient" treatment.

L 83: What morphometric measurements? These are never reported, or what are you meaning by morphometric?

L101: "hormarine and ATP-ADP, but ..."

L108-110: "significantly upregulate a host of physiological systems in order to maintain homeostasis under current pCO₂ conditions". How do we know that this metabolic profile is exclusively a response to ambient pCO₂? This could be the physiological response to this population being at a warmer temperature than it is used to. Alternately, this is just their "normal". This could alternately be said that the southern populations downregulate certain physiological systems. My point is that this sentence oversells the findings. They have distinct physiological profiles. Yes. And this is interesting.

L129-130: Nowhere in the main text is it made clear that the dissolution studies were done on empty shells. This needs to be clearly stated in the main text so as not to confuse readers. The way this reads is that growth and dissolution were measured on the same organism. Please clarify here and the methods of the main text.

L273: The respiration studies were not straightforward, put an animal in water and measure

its respiration. I think it would be useful to at least acknowledge this in the main text. Since these animals were also then used for metabolomics and the shell study understanding the recent history of their experience is valuable for consideration of the results.

Figure 1: There needs to be a label for the color bar (I assume this is water temperature...but when? What season?).

Supplementary File

Table 4 needs a better caption of what is being reported.

L60: metabolomic

L62-64: repetitive

L 169: cooled

L196-197: These references need to be formatted properly

L 235: using only...

L289-292: this sentence is long and overly complex. Could you break it up and avoid using using so many times?

L 300: analyses of shell...

L 303: Data always met the assumption..

L309: What is [ATP]? Is it adenosine tri phosphate?? If so (or if not) how was it measured?

L 347: Molluscan (I hope!)

L 376: 'southern samples'. The final AMOVA examined ...

Supplementary Table 1: I think you have switched your DIC and TA columns (DIC should vary with treatment and TA should stay the same, but that appears reversed in your table; Also you measured TA not DIC and the table says the opposite??)

L 416: for a coding region, and ...

L 423-424: Could you give us the FST value ranges?

Lischka, S., and U. Riebesell. 2012. Synergistic effects of ocean acidification and warming on overwintering pteropods in the Arctic. *Global Change Biology* 18: 3517-3528.

Manno, C., N. Morata, and R. Primicerio. 2012. *Limacina retroversa*'s response to combined effects of ocean acidification and sea water freshening. *Estuarine, Coastal and Shelf Science*

113: 163-171.

Reviewer #2 (Remarks to the Author):

The paper reports a study of the effects of ocean acidification on marine snails in a latitudinal cline and finds interesting differences. It also reports latitudinal patterns in metabolic rate, energetics and ion concentrations. These latter findings would be very important in explaining the differential impacts. However, while using sophisticated statistical tools the paper falls short of conveying a clear picture of how the different responses to CO₂ can be explained and are intertwined. Moreover, while the authors certainly investigated a lot of things and put them into one manuscript, they failed to develop how the different findings link to each other and contribute to a holistic picture. There is a significant level of superficiality involved in discussing metabolic patterns, paired with some erroneous biochemical statements, e.g., acetate is certainly not a metabolite of the citric acid cycle.

Accordingly, the paper has chosen not to integrate the insight from earlier comparative work on metabolic characteristics of marine invertebrates and fish (this and other species) in a latitudinal cline, done in the nineties and early 2000s? Considering such characteristics may have been useful in interpreting the differential impacts of CO₂ and their metabolic capacities. This should be amended. Also the latitudinal gradient chosen may not cover the full species range, and temperature variability should also be considered. It remains unclear to what extent the northern and southern populations studied really are populations at northern and southern distribution limits?

The three figures included in the material are partly different from those described in the captions and not clearly labelled?

I. 55 etc. I feel that writing lower case p for partial pressure should be abandoned. A clear term established decades ago for partial pressure is Capital italic P, see best practices guide.

I. 109, 110: very unspecific remark, see general comments

And SM I. 43. The cost aspect is somewhat speculative, I see no measurements of that.

SM Figure 6: This reads as if empty shell ATP content was correlated with other parameters?

I. 115 to 117: If this is the pattern of response it deserves explanation.

I. 122-123: acetate is no key component of the citric acid cycle.

I. 136: is a 2.4% difference significant and relevant?

I. 146 to 150: Reasoning is not really convincing and rather speculative. What does the dissolution of an isolated shell mean for the living animal?

I. 157: again extremely vague.

I. 175, 176: What does the dissolution of an isolated shell mean for the living animal?

I. 184: range-edge not demonstrated in this paper

I. 191 - 195: This is formulated as a conjecture but has not been developed based on evidence.

I. 207, 208: speculative and somehow contradictory to idea that cod-adapted population is more successful?

I. 269: how have these empty shells been obtained. The respective data need differentiated discussion of what can and cannot be applied to the living animal.

Reviewer #3 (Remarks to the Author):

Calosi et al use a multipronged approach involving population genetics, performance assays and metabolomics to test the sensitivity of snails from different latitudes to ocean acidification. The rationale is that little is known about how adaptive responses to non-temperature global change drivers will help shape future species distributions. The result demonstrate how range-edge populations are most sensitive, and the conclusion is that OA responses are likely to co-influence the future biogeography of marine ectotherm species. To the best of my knowledge these findings are novel. Although I am unsure how much this reflect OA per se or indirect responses to other effects (temperature). In general I find this study to be impressive, very interesting, well conducted and analysed and well written. I believe it would be of broad interest.

Abstract. Informative. However, the last 2 lines are a bit flat. Seem to relate to methods (trivial) rather than concept (interesting). The preceding two lines give a more crisp and profound conclusion. I am inclined to suggest deleting the last two lines and maybe add another line in the middle, providing more detail about the results.

Line 51-53 Here (and perhaps throughout) it might increase the readership and breadth if the authors did unnecessarily restrict themselves to invertebrates. Similar effects have been observed in seaweeds. See, for example, Bennett et al 2015 NCOMMS 6:10280 for a very recent and relevant example.

Line 71-72 (and in general) The rationale for the existence of latitudinal gradients in thermos-tolerance is the existence of thermo-latitudinal gradient. What is the rationale for

the proposition that similar gradients in sensitivity to pCO₂ should exist. I believe this warrants some consideration in the introduction including explain any existing or projected patterns of OA.

Line 79 Here you should mention the duration (14 d) of the high pCO₂ incubations

Line 144 Unclear. If dissolution cannot fully explain weight loss then, then what can?

Line 158 Personally I would avoid this priority claim.

Line 165 Seems the observed differences in OA sensitivity could be explained as a metabolic side-effect of thermo-regulation? Something similar has been proposed for seaweeds. See Wernberg et al 2010 *Ecol Lett* 13: 685-694.

Line 200-203 several processes could account for the loss of genotypes - selection is mentioned but it could also simply be a consequence of founder effects during post-glacial colonisation (e.g. Provan 2013 *Frontiers Biogeogr* 5: 60-66) or simply a geometric artefact of total genotype richness (Colwell & Lees 2000 *TREE* 15: 70-76). See also Hampe & Petit 2005 *Ecol Lett* 8: 461-467 for additional discussion. Expanding the argument beyond selection might make it more inclusive (or explicitly refute alternative processes if pertinent).

Fig 1 Consider if it might be acceptable to flip panels b and c to put latitude on the y-axis so as to better lign up with the map in panel a.

Fig. 2 is incredibly busy. Note sure there are any easy solutions.

Reviewers' comments:

Reviewer #1 (Remarks to the Author):

SUMMARY

This paper presents an integrated set of measurements (respiration, shell mineralogy, dissolution, metabolomics, growth) describing the response of geographically separate populations of the intertidal snail *Littorina littorea* from various locations along the European coastline to pCO₂ at one temperature (15 C). The premise was to define whether these populations show distinct responses to acidification and to explore whether biogeographic boundaries are set by sensitivity to acidification. The authors claim that their results identify population variability, both genetically and physiologically that indicate that range edges will respond differently to acidification, and with different northern and southern population responses. This is a novel set of integrated measurements that tests a hypothesis that, although I believe is accepted by the community, has not been fully articulated or rigorously explored. As such I think the perspective and measurements are of interest to both global change and evolutionary biologists, as well as policy makers and conservationists.

We appreciate the reviewer's remarks that we present << ... a novel set of integrated measurements that tests a hypothesis that has not been fully articulated or rigorously explored>>, and that our << ... perspective and measurements are of interest to both global change and evolutionary biologists, as well as policy makers and conservationists.>>. Furthermore, we appreciate that the reviewer consider our << ... a very interesting manuscript, with a lot of novel information ... >>.

I have one major concern, however, about the design and presentation of the results. All of the experiments were run at 15 C, and it is possible that the response of the northern populations, in particular, is actually an interactive stress response with temperature. As the authors already acknowledge, temperature plays a large part in shaping the distribution of marine ectotherms, and the northern populations, which are shown to have some phylogenetic distinction, may be physiologically unprepared for 15 C. To dispel this concern it would be lovely to have field data providing a range of exposure conditions for the various populations (instead of or augmenting Supplementary figure 1). This would help alleviate the concern that Tromsø snails may never/rarely experience 15 C.

Broadly I agree with the authors that taking population level responses into account is important as we move forward to a more nuanced understanding of the implications of acidification. With regards to that point, I feel that this is a very interesting manuscript, with a lot of novel information, but I am not convinced that there is not an interactive temperature effect that is being hidden by the experimental design. Arguably, however, all similar experiment that were done with temperature establishing the ranges of organisms and testing their sensitivity were similarly flawed in that they did not take into account regional changes in carbonate chemistry, salinity, oxygen, etc that vary along latitudinal gradients. Temperature is clearly one of the (if not THE) strongest driver, which is why it comes up as a red flag here. Redoing the study at local temperatures adds another set of design headaches (how are you sure that you have teased apart temperature and CO₂ if everything is at a different temperature?) and is impractical to ask the authors to do so. I don't actually think that the experimental design is irreparably flawed, I just wish for greater discussion of this point. I also need greater clarification on the methods of the exposures (detailed below) to be able to adequately understand and judge whether the respiration results are attributable to population sensitivity to pCO₂ or to temperature.

We are glad that the reviewer states <<Broadly I agree with the authors that taking population level responses into account is important as we move forward to a more nuanced understanding of the implications of acidification - and that - with regards to that point, I feel that this is a very interesting manuscript, with a lot of novel information ...>>. Furthermore, we truly appreciate the in-depth analysis and the constructive comments the reviewer has made of our work and the methodologies we used. We do agree with the reviewer that (as it is for any scientific work - as pointed out by the reviewer themselves), there are limitations that we should more clearly acknowledge and discuss.

We now do so:

(1) in the main body of the MS (Lines 259-263),
<< Maintaining all populations at common temperature conditions was an experimental requirement, whilst this may be a non-optimal temperature for some populations, it is within the range of temperatures experienced by all populations of *L. littorea* in both sea water or/and air during summer low tide (see Supplementary 1).>>

(2) Suppl. Mat. (Lines 56-68),

<<All specimens were kept in a temperature-controlled room (12:12 h L:D regime) at 15 °C. Despite the genetic distinctiveness of the populations of *L. littorina* investigated (Fig. 1a), we cannot discount the possibility that the patterns observed in the snails' physiology were generated by exposing northern populations at an experimental temperature (15 °C) that is at the upper end of their natural range of seawater temperatures experienced (particularly for the Tromsø population), but not of air temperatures (Supplementary Fig. 1). Both these Norwegian populations possess metabolomics fingerprints and metabolic rates (Fig. 1b,c) that are similar to each other, despite snails from these two populations experiencing thermal regimes that are somewhat different (Supplementary Fig. 1). This suggests that the fairly significant temperature difference between these sites ultimately does influence their metabolic responses to elevated P_{CO_2} , adding weight to the idea that the use of 15 °C did not cause bias by stressing the northernmost population.

(3) Suppl. Mat. Figure 1, please see Supplementary file.

The reviewer makes an important point about the fact that we exposed all populations to the same common temperature. They recognize that this was a methodological requirement to ensure temperature conditions were standardised and thus data (statistically) comparable, and also they do not think that this means that the experimental design is flawed. Whilst we agree that 15 °C may be a non-optimal temperature for some populations, it is within the range of temperatures experienced by most populations of *Littorina littorea* in sea water and close to 15 °C for that in Tromsø. This temperature is certainly experienced in air by all populations during summer low tides, this being relevant for an intertidal species. We have now added extra data to the Suppl. Mat. that outline more fully the temperature regimes at the sites, and following the reviewer's request [<<providing a range of exposure conditions for the various populations (..... augmenting Supplementary figure 1).>>], we now provide in the Suppl. Mat. the range of minimum and maximum temperatures in

surface sea water and atmospheric for the period 2008-2011. Time series of temperature in the intertidal environment in the sampling sites are not available, therefore we provide daily mean of sea surface temperature (source: Multi-scale Ultra-high Resolution Sea Surface Temperature by NASA <http://mur.jpl.nasa.gov/>) and daily mean of air temperature at 2 m height (source: ERA-interim from ECMWF) as proxies for the intertidal temperature. In the case of Tromsø, the average maximum sea surface temperature is about 2.5 °C lower than the reference temperature of 15 °C, however the average maximum air temperature is 16.7 °C. Considering also the diurnal variation, it is absolutely reasonable to assume that even the population at the Northern extreme of the latitudinal range experiences the temperature of 15 °C every year in the summer months both in sea water and in air during low tide.

Finally, we would also want to flag that the relatively similar response of the two Norwegian populations suggests that the fairly significant temperature difference characterising these two sites does not seem to influence their responses dramatically, adding more weight to the idea that the use of 15 °C was not causing bias by stressing the northernmost population.

We thank the reviewer for helping us improving the clarity and discussion around a very important aspect of our work. We hope that the new information provided (see Suppl. Mat. Fig. 1) <<.... help alleviate the concern that Tromsø snails may never/rarely experience 15C.>>, and that the discussion added to this point gives more balance to our work (Suppl. Mat. lines 56-68).

GENERAL COMMENTS

I would like to see an expansion of the methods associated with the respiration experiments (from which the shell and metabolomics samples are also derived). The methods of organismal exposure (in air at one of five temperatures between 15-35 C ?? Supplementary File L 127-128) are not well explained, and re-emphasize the previous concern about temperature playing a role in the observed responses of different populations. If all organisms experienced the same temperature regimes (whatever they were, possibly up to 35 C?) they would have been physiologically a larger stressor for individuals from Tromsø (I think - again details on the natural environmental variability here would help reduce the concern).

We now provide more information on the methods associated with the respiration experiments: more briefly in the main body of the MS (Lines 270-274), and more extensively in the Suppl. Mat. (Supplementary 5). Please also see details on SST and atmospheric temperatures now reported in Suppl. Mat. Fig. 1.

The second major methodological complaint I have is the use of empty shells to estimate dissolution (reported in L152 of the main text and figure 1c). It has been shown with other mollusks that dead animals dissolve much faster than live ones (Lischka and Riebesell 2012; Manno et al. 2012), which would influence the calculation of mean growth effort. I understand that the authors did not use dead, but rather empty shells, and they may believe that these differences are all due to biological effort on the part of the live organisms, but bacterial growth on empty shells and the presence of protective mucous and protein layers that are present in live individuals may also play a role. The overestimation does not, of course, account for the very large differences seen in dissolution between populations. There is clearly something different (and clearly interesting) about the shells from different regions, I just have concerns that there is no mention of the fact that this may be an overestimation.

We agree that producing an accurate measure of shell dissolution rates of living-snails is difficult and should, ideally, involve the use of differently labelled isotopes for calcium and other bivalent ions. Such techniques (e.g. those involving incubation of living snails with radioactive isotopes) were not available at Plymouth University when the experiments forming the basis of this work were undertaken, nor do they exist now in any of the institutions this group of authors is based at. In addition, with such techniques it would not have been thinkable to process the number of shells *per* population were originally processed (15 shell *per* population *per* P_{CO_2} treatment, 180 shells in total) in the period our work was carried out. Hence, we could have measured the proportional shell weight loss using either 'empty shell with an open *operculum*', or 'empty shell with a plugged *operculum*' or 'dead snails'. There are advantages and disadvantages to each of these approaches: as the reviewer correctly points out. As a substantial part of the inside of the shell of a gastropod is directly or indirectly exposed to sea water even when snails are alive, as the mantle does not cover the entire inner side of the shell and it is partly permeable, we thought it relevant to expose the inside of the shell to experimental sea water – our 'empty shell with an open *operculum* measure'. The reviewer's suggestion to recognise that this may cause a (slight?) overestimation is fair and we were happy to acknowledge it and discuss it. However, for arguments sake, we believe that other methods could cause greater overestimation (if we were to use dead snails, as pointed out by the reviewer) or underestimation (i.e. if we were to use 'empty shell with a plugged *operculum*'). We believe the method we employed is appropriate, and provides a good estimate of shell dissolution rates: particularly as the shells used came from freshly sacrificed snails where tissues were carefully removed using only plastic tools to prevent the shell to be scratch, and thus with shells' structures still intact. Finally, and most importantly, as acknowledged by the reviewer, of most importance here was the finding that << There is clearly something different (and clearly interesting) about the shells from different regions>>. We hope the reviewer is now satisfied that we recognise the possible issue of overestimating absolute values of dissolution, whilst providing a reliable measure of dissolution.

It is interesting/cool that the ATP/ADP patterns link well with the metabolic rates. There are a number of tantalizing links between the datasets that could be greater highlighted, but due to brevity constraints I recognize that not all of the potential discussion is possible. Still...

We agree that unexplored or underplayed (mechanistic) links of our data are interesting and cool ! When writing the first version of this manuscript we were cautious in not trying to answer too many questions, and actively choose to leave a more in-depth exploration of the more mechanistic links (e.g. between MO_2 and metabolomics responses) for a more detailed MS. We are however happy to add some analyses and discussion on this point, reporting about the mechanistic link between mean population metabolomics profiles and metabolic rates, and mean population metabolic rates and ATP-ADP levels (Lines 85-95, lines 125-134):

Lines 85-95,

<< In addition, the presence of a positive linear relationship between mean population metabolomic profiles and metabolic rate under control P_{CO_2} conditions ($r_5 = 0.703$, $p = 0.0184$, see Supplementary 12) suggests the existence of a clear mechanistic link between cellular and whole organism

responses (Fig. 1d). This relationship suggests a level of cellular physiology underpinning for the observed whole-organism metabolic rate adaptation along the latitudinal cline investigated (Fig. 1b). The presence of a linear positive relationship between mean population metabolic rate and ATP-ADP levels under control P_{CO_2} ($r_5 = 0.7746$, $p = 0.0409$, see Supplementary 12) further corroborate this idea, specifically suggesting the existence of a positive relationship between metabolic rate levels and cellular energy levels²²>>.

Lines 125-134,

<<In addition, the presence of a linear positive relationship between mean population metabolic rate and ATP-ADP levels under elevated P_{CO_2} ($r_5 = 0.9476$, $p = 0.0012$), suggests that the positive relationship found under control P_{CO_2} conditions between metabolic rate levels and cellular energy levels does not fundamentally change under future OA conditions across the latitudinal range of *L. littorina*. However, a positive binomial relationship was found between mean population metabolomic PC1 scores and metabolic rate under elevated P_{CO_2} ($r_5 = 0.658$, $p = 0.008$), suggesting that the mechanistic relationship observed under control P_{CO_2} conditions between the cellular and whole-organism compartments may change under future OA conditions. >>.

Supplementary file: In general I can't figure out your experimental setup. There were 120 critters per treatment and there appears to be one equilibration system per CO2 treatment that lead into 1 large holding tray per treatment, and then 30 plastic pots with one individual per pot. This means only 30 snails per treatment. Where am I missing something (L67-71)? You either had multiple equilibration systems per treatment or multiple holding trays? This also plays into confused math in line 127 where there is a mention of multiple experimental tanks??? How does that play back into the trays and pots?

We appreciate that the description of the experimental set up may have lacked some detail and have improved the clarity of this description. Please refer to Supplementary 5.

Supplementary file section 9.1: what statistical program did you use? When you say that only the best fit model are shown are you referring section 12? This part of the statistics is not very clear to me.

The software used to conduct the analyses was SPSS v19. This information is now reported in the methods section of the main body of the MS, and in Supplementary 12.

Our Supplementary Materials are clearly divided between 'Additional Details for Materials and Methods' (sections 1-10), 'Additional Results' (sections 10-13 plus supplementary tables and figures) and 'References'. In section 9.1 we explain the approach we used. This is pretty standard and we find simple enough for reader to understand the exercise we carried out, not requiring any further explanation or clarification, if the reviewer could give us more details on what they think it is not clear further comments would be extremely welcome. In relation to section 12 here we report the statistical details for significant relationship reported in the main text and represented in the figures.

Supplementary file: The three temperature regimes defined in the main text L 78-79 are not defined in the supplementary data before their first mention (L 314) and are then re-defined with

different names (L 373-376). Consistency of naming and a reintroduction of the idea before the metabolomics section of the supplementary file would be useful.

This issue is now resolved, and we consistently define the three regions always referring to the 'formula' used in the main body of the MS: << ... three distinctive climatic regions: i) warm-temperate; ii) cold-temperate; and iii) sub-polar, ... >>. We use these terms in the Suppl. Mat. right at the beginning of the methods Suppl. Mat. 1 (Lines 28-29) before the metabolomics section in the Suppl. Mat. as suggested by the reviewer. We also introduce this concept right at the beginning of the main body MS (Lines 49 and 76-77) and use these terms consistently throughout the MS.

Supplementary file: The way that the shell mineralogy results are presented needs some polishing (L 528-541). My confusion has to do with the use of the words lowest/highest and least/most affected as if they were interchangeable. Least effected might be better explained as "showed the least change"? The question is whether you are reporting the absolute values (highest/lowest) or the sensitivity (more or less change). The wording needs cleaned up. The figures also need some clarification (and better captions). I think that the * means that there is an effect of pCO₂ overall, not that there is a bonferoni post-hoc significance of pCO₂ at that particular location, but I could be wrong. The scale of the y axis makes it hard to determine whether there are differences in some of the graphs.

We report the absolute value and discuss the % change in mineral content. We have followed the review's suggestion and polished this section accordingly. Regarding the scale of the y axis, we do feel important to show the full scale including the zero values, as changes whilst significant are small. Changing the scale of the y axis would bias the readers' perception, in thinking changes are larger then they are, and as we do not wish to be misleading we have decided to maintain the scale as it is. We have revised the caption, ensuring they are sufficiently clear. The symbol '*' does mean that there is an effect of pCO₂ on mean concentration or ratio of ions at the population level.

Supplementary file: In generally you need greater consistency in labeling. Sometimes samples are identified by latitude, sometimes by name of site, sometimes by code (country A, B).

We have removed all codes, and minimised the used of site names. In general we use site names when we wish to identify specific populations, and use latitude for regression analyses. Only where required we have standardised labelling of populations across the MS as: Vig-Vigo, IdR-Île de Ré, Ros-Roscoff, Mil-Millport, Trd-Trondheim, Trm-Tromsø.

Supplementary file: It would be nice if the tables and graphs were always pointed to throughout the text of the supplementary file so that we know which thing is being referenced.

We have taken care to ensure that tables and figures are pointed to throughout the text of the Suppl. Mat. to better help the reader in finding the relevant information at the right point. However, in some cases the information provided in the Suppl. Mat. is linked to the text in the main body of the MS, and thus this explains why there is not a reference within the suppl. text in this section.

SPECIFIC COMMENTS

Main manuscript

L 81 and L 252-253: the authors are inconsistent with their description of the pCO₂ treatments. Better I think would be to list the achieved levels rather than the desired levels (~420 and 1000 from the supplementary file). Although they did not get as low as they would have liked, 420 is not unreasonable at all for a modern day intertidal organism and is quite reasonable as an "ambient" treatment.

We now more consistently referred to the obtained pCO₂ (P_{CO_2} following Rev. #2 request) values.

L 83: What morphometric measurements? These are never reported, or what are you meaning by morphometric?

This was rectified. It now reads << Total body mass measurements ...>>.

L101: "hormarine and ATP-ADP, but ..."

Amended.

L108-110: "significantly upregulate a host of physiological systems in order to maintain homeostasis under current pCO₂ conditions". How do we know that this metabolic profile is exclusively a response to ambient pCO₂? This could be the physiological response to this population being at a warmer temperature than it is used to. Alternately, this is just their "normal". This could alternately be said that the southern populations downregulate certain physiological systems. My point is that this sentence oversells the findings. They have distinct physiological profiles. Yes. And this is interesting.

Our phrasing was misleading; we meant that they are different, as demonstrated by the fact that under control conditions for P_{CO_2} they show to have very different metabolomics fingerprints. We did not mean to say that they are upregulating to keep homeostasis at control conditions when compared to other populations. Rather we wanted to say that they show differences in their metabolomes also at control conditions. We do not believe that temperature is here the driving force for the change in metabolomic profiles.

Whilst we agree that 15 °C may be a non-optimal temperature for some populations, it is within the range of temperatures experienced by most populations of *Littorina littorea* in sea water and close

to 15 °C for that in Tromsø, and certainly in air by all populations during summer low tides. We have now added extra data to the Suppl. Mat. that outline more fully the temperature regimes at the sites, and following the reviewer's request [<<providing a range of exposure conditions for the various populations (..... augmenting Supplementary figure 1).>>], we now provide in the Suppl. Mat. the range of minimum and maximum temperatures for the period 2008-2011. Time series of temperature in the intertidal environment in the seven sampling sites are not available, therefore we provide daily mean of sea surface temperature (source: Multi-scale Ultra-high Resolution Sea Surface Temperature by NASA <http://mur.jpl.nasa.gov/>) and daily mean of air temperature at 2 m height (source: ERA-interim from ECMWF) as proxies for the intertidal temperature. In the case of Tromsø, the average maximum sea surface temperature is about 2.5 °C lower than the reference temperature of 15°C, however the average maximum air temperature is 16.7°C. Considering also the diurnal variation, it is absolutely reasonable to assume that even the population at the Northern extreme of the latitudinal range experiences the temperature of 15°C every year in the summer months both in sea water and in air during low tide.

Finally, we would also want to flag that the relatively similar response of the two Norwegian populations suggests that the fairly significant temperature difference between these sites does not seem to influence their responses dramatically, adding more weight to the idea that the use of 15 °C was not causing bias by stressing the northernmost population. We now acknowledge in the Supplementary 1 <<All specimens were kept in a temperature-controlled room (12:12 h L:D regime) at 15 °C. Despite the genetic distinctiveness of the populations of *L. littorina* investigated (Fig. 1a), we cannot discount the possibility that the patterns observed in the snails' physiology were generated by exposing northern populations at an experimental temperature (15 °C) that is at the upper end of their natural range of seawater temperatures experienced (particularly for the Tromsø population), but not of air temperatures (Supplementary Fig. 1). Both these Norwegian populations possess metabolomics fingerprints and metabolic rates (Fig. 1b,c) that are similar to each other, despite snails from these two populations experiencing thermal regimes that are somewhat different (Supplementary Fig. 1). This suggests that the fairly significant temperature difference between these sites ultimately does influence their metabolic responses to elevated P_{CO_2} , adding weight to the idea that the use of 15 °C did not cause bias by stressing the northernmost population.>>.

L129-130: Nowhere in the main text is it made clear that the dissolution studies were done on empty shells. This needs to be clearly stated in the main text so as not to confuse readers. The way this reads is that growth and dissolution were measured on the same organism. Please clarify here and the methods of the main text.

We now indicate that dissolution trials were carried out on individual 'empty' shells.

L273: The respiration studies were not straightforward, put an animal in water and measure its respiration. I think it would be useful to at least acknowledge this in the main text. Since these

animals were also then used for metabolomics and the shell study understanding the recent history of their experience is valuable for consideration of the results.

We have specified that metabolic rate measures were carried out in sea water at 15 °C, following exposure to a range of current and future temperatures in air in an attempt to measure metabolic performances of snail populations under summer low tide: namely following the time snails experience the greatest physiological challenge. More and clearer information is now provided in the main body of the MS (Lines 270-274), and more extensively in the Suppl. Mat. (Supplementary 5). However, whilst we agree that the recent history is relevant for the interpretation of metabolomics responses, and this is now better integrated in the MS, we do not believe that the 'short' exposure history during respiration trial would have had a significant impact on shell mineralogy.

Figure 1: There needs to be a label for the color bar (I assume this is water temperature...but when? What season?).

Do to a strict word limits for captions (100 words *per* caption maximum) we cannot add this information to the legend, but we do recognise this is extremely important and we try now to provide as much information as possible in the figure itself. The reviewer can now see that the colour bar represents a scale of 'Average annual mean of seawater temperature (°C) simulated by the biogeochemical model POLCOMS-ERSEM 1980-2004'.

All changes in the Supplementary files were implemented

Supplementary File

Table 4 needs a better caption of what is being reported.

The caption was improved and symbols are now explained.

L60: metabolomic

Amended in all cases where 'metabolomic' is used as an adjective .

L62-64: repetitive

We agree but the Suppl. Mat. must be a stand-alone document, and thus we need to leave this information here.

L 169: cooled

Modified.

L196-197: These references need to be formatted properly

Amended.

L 235: using only...

Amended.

L289-292: this sentence is long and overly complex. Could you break it up and avoid using using so many times?

The sentence was separated into two, and phrasing improved.

L 300: analyses of shell...

Amended.

L 303: Data always met the assumption..
Modified.

L309: What is [ATP]? Is it adenosine tri phosphate?? If so (or if not) how was it measured?
The acronym is now explained, and it is specified that technical details are below in the text.

L 347: Molluscan (I hope!)
Indeed! This has been corrected.

L 376: 'southern samples'. The final AMOVA examined ...
Modified.

Supplementary Table 1: I think you have switched your DIC and TA columns (DIC should vary with treatment and TA should stay the same, but that appears reversed in your table; Also you measured TA not DIC and the table says the opposite??)
We have switched TA and DIC values, and now indicate that we did in fact measure TA and calculated DIC.

L 416: for a coding region, and ...
Modified.

L 423-424: Could you give us the FST value ranges?
FST values range was added to Supplementary Materials.

Lischka, S., and U. Riebesell. 2012. Synergistic effects of ocean acidification and warming on overwintering pteropods in the Arctic. *Global Change Biology* 18: 3517-3528.
Manno, C., N. Morata, and R. Primicerio. 2012. *Limacina retroversa*'s response to combined effects of ocean acidification and sea water freshening. *Estuarine, Coastal and Shelf Science* 113: 163-171.

These references were added to the main text.

Reviewer #2 (Remarks to the Author):

The paper reports a study of the effects of ocean acidification on marine snails in a latitudinal cline and finds interesting differences. It also reports latitudinal patterns in metabolic rate, energetics and ion concentrations. These latter findings would be very important in explaining the differential impacts.

We are very glad that the reviewer finds our results interesting, and sees value in our macrophysiological approach to help define sensitivity to ocean acidification at the population level.

However, while using sophisticated statistical tools the paper falls short of conveying a clear picture of how the different responses to CO₂ can be explained and are intertwined. Moreover, while the authors certainly investigated a lot of things and put them into one manuscript, they failed to develop how the different findings link to each other and contribute to a holistic picture. There is a significant level of superficiality involved in discussing metabolic patterns, paired with some erroneous biochemical statements, e.g., acetate is certainly not a metabolite of the citric acid cycle.

We are disappointed that this referee does not feel that the overall picture we convey is clear and holistic, and that we have been superficial in our discussion of metabolic patterns. We would argue that, whilst our MS does not single handily focus on mechanistic aspects of the response of *Littorina littorea* to ocean acidification, it does integrate elements of physiology, life history, biogeography and evolution with the aim (for first time within a structured and rigorous manner as Reviewers #1 acknowledge) to define the sensitivity of multiple populations of *Littorina littorea* living along an environmental gradient to future ocean acidification conditions. Our aim was to make our integrative and multidisciplinary work accessible to a broader audience. This necessitated a compromise, i.e. provide a lesser depth of the discussion of specific aspects of our work to focus on the synthesis of the different topics covered. We feel we have achieved our goal, and produced a scientifically sound and useful study, of interest for and accessible to a broad audience (as Reviewers #1 and #3 state). Nonetheless, we agree with Reviewer #2 that exploring the mechanistic links between functions and compartments is extremely interesting, and we have added the investigation of the relationship between mean population metabolomics profiles and metabolic rates, and mean population metabolic rates and ATP-ADP levels (Lines 85-95, lines 125-134, see below). At the same time, we are sure the reviewer appreciates that the space for references is substantially limited, and that a detailed investigation of the physiological implications and mechanistic linkages will have to be deferred to a future paper intended for a specialist journal.

Lines 85-95,

<< In addition, the presence of a positive linear relationship between mean population metabolomic profiles and metabolic rate under control P_{CO_2} conditions ($r_s = 0.703$, $p = 0.0184$, see Supplementary 12) suggests the existence of a clear mechanistic link between cellular and whole organism responses (Fig. 1d). This relationship suggests a level of cellular physiology underpinning for the observed whole-organism metabolic rate adaptation along the latitudinal cline investigated (Fig. 1b). The presence of a linear positive relationship between mean population metabolic rate and ATP-ADP levels under control P_{CO_2} ($r_s = 0.7746$, $p = 0.0409$, see Supplementary 12) further corroborate this idea, specifically suggesting the existence of a positive relationship between metabolic rate levels and cellular energy levels²²>>.

Lines 125-134,

<<In addition, the presence of a linear positive relationship between mean population metabolic rate and ATP-ADP levels under elevated P_{CO_2} ($r_5 = 0.9476$, $p = 0.0012$), suggests that the positive relationship found under control P_{CO_2} conditions between metabolic rate levels and cellular energy levels does not fundamentally change under future OA conditions across the latitudinal range of *L. littorina*. However, a positive binomial relationship was found between mean population metabolomic PC1 scores and metabolic rate under elevated P_{CO_2} ($r_5 = 0.658$, $p = 0.008$), suggesting that the mechanistic relationship observed under control P_{CO_2} conditions between the cellular and whole-organism compartments may change under future OA conditions. >>.

Finally, we have rectified the erroneous statement made, as pointed out by the reviewer. We are grateful to the reviewer for reporting this. This has been corrected as follows Some metabolomic differences for warm-temperate sites were also highlighted by PLS-DA analyses (Fig. 2d-i) under elevated P_{CO_2} : the Île de Ré population showed a significant decrease in acetate, an essential ingredient for the formation of the co-enzyme acetyl-co-A, which enables energy production; the Vigo population showed a significant decrease in ATP (or ADP) concentrations, which is of fundamental importance for central metabolism²⁷. >>

Accordingly, the paper has chosen not to integrate the insight from earlier comparative work on metabolic characteristics of marine invertebrates and fish (this and other species) in a latitudinal cline, done in the nineties and early 2000s? Considering such characteristics may have been useful in interpreting the differential impacts of CO2 and their metabolic capacities. This should be amended.

We start our MS by stating << Populations of the same species from different climatic regions often differ in their ability to withstand altered environmental conditions^{1,2}. In the marine realm, diversification of populations along thermo-latitudinal gradients has been demonstrated for traits related to thermal physiology in several taxonomic groups¹⁻⁴, >>. Three out of four of the references we used in this section are reviews (here below in bold) on comparative physiology along gradients (i.e. macrophysiology) partly or completely focussing on marine systems. These papers review and synthesise existing knowledge on the <<... metabolic characteristics of marine invertebrates and fish (this and other species) in a latitudinal cline...>>, including classical studies from the '90s and '00s, and earlier. In addition, many other references relevant to this field can be found throughout the MS. Given the limitation of 50 references we feel that we have integrated insights from earlier work for as much as possible given these constraints.

On another note, and linked to our response above to the reviewer's request to produce a more in-depth discussion of the physiological implications of our findings, we hope the reviewer can appreciate that integrating aspects of life history, evolution and physiology within a limit of 50 references puts a great deal of pressure on reference selection. Nonetheless, in an attempt to respond to both Reviewer #2 and Reviewer #1 requests to improve our discussion on the implications for metabolic rate responses to elevated pCO_2 , we have now added the investigation of the relationship between different physiological parameters, and have improved the depth of our

discussion particularly for our physiological data in multiple points in the main body of the MS (see for example Lines 85-95, lines 125-134).

1. Gaston, K. J. et al. **Macrophysiology: A Conceptual Reunification**. *Am. Nat.* **174**, 595-612 (2009).
2. Bozinovic, F., Calosi, P. & Spicer, J. I. **Physiological Correlates of Geographic Range in Animals**. *Annu. Rev. Ecol. Evol.* **41**, 121-149 (2010).
3. Dam, H. G. **Evolutionary adaptation of marine zooplankton to global change**. *Annu. Rev. Mar. Sci.* **5**, 349-370 (2013).
4. Bennett, S. et al. **Central and rear-edge populations can be equally vulnerable to warming**. *Nat. Comm.* **6**:10280 (2015).

Also the latitudinal gradient chosen may not cover the full species range, and temperature variability should also be considered. It remains unclear to what extent the northern and southern populations studied really are populations at northern and southern distribution limits?

We are confident that the populations used are representative of *Littorina littorea* close to the southern and northern edges of its range. Firstly, we canvassed the opinions of experienced marine ecologists and physiologists from Portugal, Spain, France, UK, Norway, Iceland and Russia about the known distribution of the species. Our chosen populations were a pragmatic compromise between what was feasible within the logistical and time constraints of the project. Hence with regard to its northernmost range, *Littorina littorea* is found in colder environments in the White Sea in Russia, but this location is further south than those in northern Norway, thus 'breaking' the south-north latitudinal range we selected. For the southern populations, whilst there are records of the species in Portugal, we were unable to locate any of these (despite asking for help from local scientists) and there is a *consensus* that populations in the south of Portugal may have disappeared: here however over-harvesting (which only occur in Portugal) of this species may be the/an important cause for such disappearance. There is apparently a residual population in the Canary Islands (see also Panova et al. 2015), but as this is considered a biogeographic relict we decided not to consider it for our study. Nevertheless, we appreciate this point made by the referee and have now referred to the range edge populations as being 'towards' the range edge rather than suggesting that the population used was 'at' the edge of the range.

Finally, we would like to point out that our study is to date the most extensive and rigorous comparison of multiple populations along a latitudinal and environmental gradient within the context of ocean acidification, and here we stress more than two populations, which is an essential methodological requirement if we are to assess the evolutionary implications of living under different climatic regions.

The three figures included in the material are partly different from those described in the captions and not clearly labelled?

Captions were rectified to match the figures.

I. 55 etc. I feel that writing lower case p for partial pressure should be abandoned. A clear term established decades ago for partial pressure is Capital italic P, see best practices guide.

The symbol $p\text{CO}_2$ was modified to P_{CO_2} as requested. However, we would like to point out that in relation to the Best Practices Guide the use of P_{CO_2} is well accepted for organisms acid-base status, whilst for seawater CO_2 partial pressure multiple symbols are suggested $p(\text{CO}_2)$ appearing the favoured one. We personally would favour $p\text{CO}_2$ but accept the reviewer request.

Table: Main parameters describing the physicochemistry of seawater and body fluids and their differences in the fields of marine chemistry and physiology. The notation and units used in this guide are also shown. Alternate notations or units are given in parentheses.

Parameter	Marine chemistry		Physiology	
	Notation	Unit	Notation	Unit
pH ⁽¹⁾	Total scale	-	NBS or NIST scale ⁽²⁾	-
Partial pressure of CO_2	$p(\text{CO}_2)$ ($p\text{CO}_2$, P_{CO_2} , $p(\text{CO}_2)$)	μatm	P_{CO_2}	kPa (mm Hg, Torr, μatm)

I. 109, 110: very unspecific remark, see general comments

This section was modified, together with the prior discussion of the metabolomics data also following request from Reviewer #1 and it now reads as << Sub-polar (Norwegian) populations had the most divergent metabolomic profiles (Fig. 2a,b) with lower concentrations of homarine and ATP-ADP but higher concentrations of leucine, valine, glutamate, glutamine and formate compared with other sites (for formate and ATP-ADP see Fig. 2k and 2n). Amino acids are used extensively in energy metabolism^{23,24} and can function as an alternative carbon source in the citric acid cycle when ATP has been exhausted. The ratios of adenylates and amino acids seen here could indicate changes that have occurred in the underlying metabolic machinery of northern populations that have allowed snails to adapt metabolically to cooler climates (Fig. 1b). Furthermore, in marine molluscs, homarine and branched chain amino acids (such as leucine and valine) also underpin processes related to acid-base balance, and immune function, which are intertwined with energy metabolism. By comparison, glutamate has been highlighted as a possible excitatory neurotransmitter in molluscs²⁵. Thus, our data suggest that, compared with more southern populations, sub-polar snails significantly upregulate a host of physiological systems, including metabolism and acid-base balance and immune and neurological function, to help maintain homeostasis (see Supplementary 1 for further discussion). Please also see above our rationale for not being able to go into any further detailed discussion of some important aspects of our findings stemming from our work.

And SM I. 43. The cost aspect is somewhat speculative, I see no measurements of that.

We have toned down this statement, and now state that << ... differences **may** exist in the way shells are built and in the costs associated with mineralisation ... >>.

SM Figure 6: This reads as if empty shell ATP content was correlated with other parameters?

We thank the reviewer for helping us improving the clarity of the legend in Figure 6.

I. 115 to 117: If this is the pattern of response it deserves explanation.

We have added an interpretation to the pattern found: <<These results suggest that populations' levels of metabolic adaptation to prevalent regional environmental conditions lead to different sensitivity to elevated P_{CO_2} conditions, with populations found towards the range edges being more sensitive. Similarly, a greater degree of sensitivity to elevated P_{CO_2} has been proposed for stenotherms compared to eurytherms, based on their different levels of thermal physiological adaptation²⁶.>>.

I. 122-123: acetate is no key component of the citric acid cycle.

This has been corrected as follows << Some metabolomic differences for warm-temperate sites were also highlighted by PLS-DA analyses (Fig. 2d-i) under elevated P_{CO_2} : the Île de Ré population showed a significant decrease in acetate, an essential ingredient for the formation of the co-enzyme acetyl-co-A, which enables energy production; the Vigo population showed a significant decrease in ATP (or ADP) concentrations, which is of fundamental importance for central metabolism²⁷.>>.

I. 136: is a 2.4% difference significant and relevant?

We have removed this statement and the section, which we understand was somewhat misleading, now reads as << Under current P_{CO_2} conditions there was no relationship between growth rate and latitude (Fig. 3a).>>.

I. 146 to 150: Reasoning is not really convincing and rather speculative. What does the dissolution of an isolated shell mean for the living animal?

We respectfully disagree with the reviewer that our <<Reasoning is not really convincing and rather speculative.>>. We remain convinced that the argument we put forward, corroborated by relevant information in the main body of the MS, figures and table, and Suppl. Mat., is solid and reliable.

As for the second part of the commentary: << What does the dissolution of an isolated shell mean for the living animal?>>.

There is not an easy way to produce an accurate measure of living-snails shell dissolution rates, which does not involve differently labelled isotopes for calcium and other bivalent ions. Such techniques (e.g. those involving incubation of with radioactive isotopes) were not available at Plymouth University when the experiments forming the basis of this work were undertaken, nor do they exist now in any of the institutions this group of authors is based at. In addition, even with such techniques it would not have been practicable to process the number of specimens *per* population we processed (240 shells) within the duration of the project. Thus we were left (as for most researchers) with the measurement of the proportional shell weight loss using either 'empty shell with an open *operculum*', or 'empty shell with a plugged *operculum*' or 'dead snails'. Any of these choices could have been plausible, each with its own pros and cons, and each producing estimates (as Reviewer #1 correctly points out) of shell dissolution. As a substantial part of the inside of the shell of a gastropod is directly or indirectly exposed to sea water even when snails are alive, as the mantle does not cover the entire inner side of the shell, we thought it relevant to also expose the inside of the shell to experimental sea water. As a consequence, we decided to use 'empty shell with an open *operculum*'. Reviewer #1 suggests we recognise that this could cause a (slight) overestimation, and we are happy to do. However, as we argue above, other methods could cause greater overestimation (if we were to use dead snails, as pointed out by Reviewer #1) or underestimation (i.e. if we were to use 'empty shell with a plugged *operculum*'). We believe the method we employed is appropriate, and provides a good estimate of shell dissolution rates: particularly as the shells used came from freshly sacrificed snails where tissues were carefully removed with plastic or Teflon-coated tools to prevent the shell to be scratch, and thus with shells' structures still intact. What we measured in 'living' snails is the result of their net growth *versus* passive dissolution (as % change in weight) over 14 days of exposure to control and elevated P_{CO2} conditions. What we measured in 'empty' shells is the passive dissolution (again as % change in weight). The difference (delta) between these mean values *per* population provides the actual growth effort. We hope the reviewer will recognise that we attempted to balance the possible issue of overestimating absolute values of dissolution, whilst providing a reliable measure of dissolution.

I. 157: again extremely vague.

We would again suggest that our statement here (and reproduced below) is clear, certainly concise as demanded by the journal style, and our argumentation stem from corroborated evidence, and the discussion is supported by relevant references. << Nonetheless, the increased shell dissolution rates, as a result of the exposure to sea water with low carbonate saturation states²⁹, appear to eclipse any potential increase in calcium carbonate deposition resulting in a negative net calcification rate (Fig. 3b) - a mechanism that could explain the negative growth reported under OA conditions. Calculations of population net growth, showed that this was 4-8 times greater in sub-polar populations compared with the other two climatic regions (Fig. 3c), suggesting that the former up-regulate shell mineralization to compensate for extreme levels of shell dissolution as response to exposure to corrosive waters³² and, likely, with energetic and metabolic implications^{22,28,29,33}.

I. 175, 176: What does the dissolution of an isolated shell mean for the living animal?

Please see our response to the same question above.

I. 184: range-edge not demonstrated in this paper

We have improved the clarity of the information provided originally on the biogeography of *Littorina littorea* in the main body of the MS, also adding a relevant reference already given below in the MS. In addition, we now clearly point to the additional information given in the Suppl. Mat, which is further supported by relevant references.

I. 191 - 195: This is formulated as a conjecture but has not been developed based on evidence.

Here our statement is based on our novel results, and we feel it is unfair to state our 'conjecture' is not based on evidence as we provide and link hard experimental evidence. We state in the main body of the MS that central populations are those that show the lowest loss in growth under elevated P_{CO_2} conditions, and the only one showing increased metabolic rates. Further, we provide a mechanistic link between population metabolic capacity and their future biogeography, our reasoning further supported by a number of relevant references on elevated P_{CO_2} . Indeed, we would argue that these conjectures are the type of synthesis and discussion that this referee suggested was lacking from the MS.

I. 207, 208: speculative and somehow contradictory to idea that cold-adapted population is more successful?

Here we feel that we are in line with what the reviewer is suggesting. We are not stating that cold-adapted populations are going to be the most successful, but that, on the contrary, they are the most at risk of extension, and could be replaced by central populations. Here, based on genetic, life-history and physiological data we obtained, we attempt to make predictions on the future biogeography of *Littorina littorea*, and in doing so we must logically 'speculate' on potential shifts, restrictions and expansions of its future geographical range. In order to support our logical reasoning using our novel hard evidence, as well as existing knowledge, on this species and the issues we tackle, we provided a discussion based on populations' regional responses, which is what we found to be the strongest pattern of response to future elevated P_{CO_2} .

I. 269: how have these empty shells been obtained. The respective data need differentiated discussion of what can and cannot be applied to the living animal.

We added << from a set of freshly sacrificed snails (n = 30) from each population>>. We have ensured to improve clarity to what apply to living snails and empty shells.

Reviewer #3 (Remarks to the Author):

Calosi et al use a multipronged approach involving population genetics, performance assays and metabolomics to test the sensitivity of snails from different latitudes to ocean acidification. The rationale is that little is known about how adaptive responses to non-temperature global change drivers will help shape future species distributions. The result demonstrate how range-edge populations are most sensitive, and the conclusion is that OA responses are likely to co-influence the future biogeography of marine ectotherm species. To the best of my knowledge these findings are novel. Although I am unsure how much this reflect OA per se or indirect responses to other effects (temperature). In general I find this study to be impressive, very interesting, well conducted and analysed and well written. I believe it would be of broad interest.

We thank the reviewer for stating that our study is <<... impressive, very interesting, well conducted and analysed and well written.>>, and to recognise that << ... it would be of broad interest.>>.

Abstract. Informative. However, the last 2 lines are a bit flat. Seem to relate to methods (trivial) rather than concept (interesting). The preceding two lines give a more crisp and profound conclusion. I am inclined to suggest deleting the last two lines and maybe add another line in the middle, providing more detail about the results.

We have taken up the reviewer's comment and in the abstract we have: (1) added more information on the results, (2) modified the punch line.

Line 51-53 Here (and perhaps throughout) it might increase the readership and breadth if the authors did unnecessarily restrict themselves to invertebrates. Similar effects have been observed in seaweeds. See, for example, Bennett et al 2015 NCOMMS 6:10280 for a very recent and relevant example.

We thank the reviewer for giving us the opportunity to further broaden the scope of our paper. We have taken up the suggestion and added the suggested reference.

Line 71-72 (and in general) The rationale for the existence of latitudinal gradients in thermotolerance is the existence of thermo-latitudinal gradient. What is the rationale for the proposition that similar gradients in sensitivity to pCO₂ should exist. I believe this warrants some consideration in the introduction including explain any existing or projected patterns of OA.

There is indeed a pH/ P_{CO_2} and $[\text{CO}_3^{2-}]$ latitudinal gradient, as shown by Orr et al. 2005 'Anthropogenic ocean acidification over the twenty-first century and its impact on calcifying organisms' *Nature* 437,

681-686. However, here the point that we are making is that extant level of 'local' and even more so 'regional' adaptation drives populations' responses to future OA. This suggests that future levels of OA could mediate the (primarily) temperature-driven shifts in species distributions. We do believe that thermal adaptation plays a major role in defining the different levels of sensitivity observed in the populations investigated, but we are aware that other factors may help defining extant levels of local/regional adaptation (incl. pH, P_{CO_2} , P_{O_2} , salinity, $[CO_3^{2-}]$, and many others in isolation and interacting). The exercise to dissect the environmental factors defining local and regional adaptation will be part of future investigations that requires heavy duty model selection exercises for all possible combinations of environmental factors in isolation and as interactions (as we conducted for a relatively small set of data on physiological windows of diving beetle species with different biogeography, see Calosi et al. 2010 *J. Anim. Ecol.*). Such exercise goes well beyond the scope of the present paper, but it is something we intend to undertake in the future.

Line

Line 158 Personally I would avoid this priority claim.

The statement from line 158 to line 160 was removed altogether.

Line 165 Seems the observed differences in OA sensitivity could be explained as a metabolic side-effect of thermo-regulation? Something similar has been proposed for seaweeds. See Wernberg et al 2010 *Ecol Lett* 13: 685-694.

We agree, and have modified the text accordingly and added the suggested reference.

Line 200-203 several processes could account for the loss of genotypes - selection is mentioned but it could also simply be a consequence of founder effects during post-glacial colonisation (e.g. Provan 2013 *Frontiers Biogeogr* 5: 60-66) or simply a geometric artefact of total genotype richness (Colwell & Lees 2000 *TREE* 15:70-76). See also Hampe & Petit 2005 *Ecol Lett* 8:461-467 for additional discussion. Expanding the argument beyond selection might make it more inclusive (or explicitly refute alternative processes if pertinent).

We agree that other genetic mechanisms than selection alone will play an important role in defining species future biogeography under the ongoing global environmental change. However, we feel that our evidence cannot support an in detail discussion of these other (potential) future mechanisms. This said, we acknowledge the importance of populations' post-glacial history (mentioning a key reference from Panova et al. 2015) in potentially causing physiological maladaptation in sub-polar (northern range) populations.

Fig 1 Consider if it might be acceptable to flip panels b and c to put latitude on the y-axis so as to better lign up with the map in panel a.

We agree that on one level this solution would make the figures more intuitive, but on the other hand it would make them diverge from typical figures with the determining factor (here 'latitude') on the x-axis. As here we mean causation, flipping Figures 1b and 1c would make them less rigorous, and thus we prefer to keep these figures as they are.

Fig. 2 is incredibly busy. Note sure there are any easy solutions.

We agree that Figure 2 is busy, despite our attempts to reduce all relevant information to the minimum. However, we feel that all information in Figure 2 is relevant for readers to appreciate our findings, and we feel that it is important these results are represented in the main body of the MS.

REVIEWERS' COMMENTS:

Reviewer #1 (Remarks to the Author):

The authors have done a very nice job of addressing most of my concerns. I appreciate the effort made to further discuss the temperature range and sensitivity of the populations. There were a few minor edits that remain and one point I felt was not entirely addressed.

Remaining Concerns:

I still don't understand L 161-163 of the supplement. Were individuals from each population exposed to the variable air temperatures documented, or were these differences in air temperature different among populations? In the response reviewers the authors state that the measurements were made "following exposure to a range of current and future temperatures in air in an attempt to measure metabolic performances of snail populations under summer low tide". From this it seems that the air temperature was different based on different populations. This needs to be stated clearly as it may influence the resulting metabolic rates and metabolomics.

Minor Edits:

In the amendment to lines S,56-68 I believe the authors meant to state that these sites "ultimately does NOT influence...", although I may be misunderstanding their meaning.

Line 70, "uses data FROM population genetics..."

Need a figure legend for Figure S1. It looks like the text was there but unreadable in my version (L 51-33).

SL569 Tromso

I did not intend for the authors to be forced to include the citations I included (particularly with the limit on references cited) – these were just references added to support the idea that shell dissolution rates change in live versus dead animals. I apologize for the confusion.

Reviewer #2 (Remarks to the Author):

The authors have made an effort to improve their manuscript and have been successful, within limits. I have an issue with broadscale, sometimes trivial statements on patterns seen in the omic data which do not necessarily deepen our knowledge but this is a more general point which authors may find hard to overcome. However, due to some bias with their own literature on macrophysiology in a latitudinal cline, their discussion of metabolism still ignores the rich literature on hypoxia adaptation in invertebrates, the detailed knowledge about differential mitochondrial densities and metabolic mechanisms and regulation in animal tissues in a latitudinal cline, the use of anaerobic metabolism including

acetate in invertebrates at thermal limits etc. This should still be alleviated. Otherwise I am happy with the revisions and think the paper makes a valuable contribution to understanding the diversity of responses to ocean acidification in animal populations, identifying the position of the population within such range as one of the factors shaping that response diversity.

Reviewer #3 (Remarks to the Author):

The authors have responded comprehensively on all queries and criticisms raised by the three reviewers. I have nothing further to add.

Two comments on the figures for the authors (and the editor) to consider:

Figure 1 is, in my opinion disconnected across panels with the current layout defying the purpose of grouping these panels. While I understand the authors' reluctance to display latitude on the y-axes of figure 1 panels b-d as this is not the convention to infer mechanism I think there is still a case for this to maximise the visual impact of the figure. I am sure the readers will be smart enough to understand if a succinct explanation/justification is added to the caption or methods. Similarly, while I understand that scale bars normally go from low to high, I think it would be more logical in Fig1a if the scale bar followed the latitudinal gradient from warm to cool.

Figure 2 has a lot of wasted blank space around all the panels and could be condensed somewhat.

REVIEWERS' COMMENTS:

Reviewer #1 (Remarks to the Author):

The authors have done a very nice job of addressing most of my concerns. I appreciate the effort made to further discuss the temperature range and sensitivity of the populations. There were a few minor edits that remain and one point I felt was not entirely addressed.

Remaining Concerns:

I still don't understand L 161-163 of the supplement. Were individuals from each population exposed to the variable air temperatures documented, or were these differences in air temperature different among populations? In the response reviewers the authors state that the measurements were made "following exposure to a range of current and future temperatures in air in an attempt to measure metabolic performances of snail populations under summer low tide". From this it seems that the air temperature was different based on different populations. This needs to be stated clearly as it may influence the resulting metabolic rates and metabolomics.

RE: We thank the reviewer for the positive remarks on the quality of our review. We have clarified this point in the main text of the MS (now lines 437-459).

Minor Edits:

In the amendment to lines S,56-68 I believe the authors meant to state that these sites "ultimately does NOT influence...", although I may be misunderstanding their meaning.

RE: amended.

Line 70, "uses data FROM population genetics..."

RE: modified.

Need a figure legend for Figure S1. It looks like the text was there but unreadable in my version (L 51-33).

RE: amended, now in main text (Fig. 2).

SL569 Tromso

RE: amended.

I did not intend for the authors to be forced to include the citations I included (particularly with the limit on references cited) – these were just references added to support the idea that shell dissolution rates change in live versus dead animals. I apologize for the confusion.

RE: We are grateful for this clarification and we have made an effort to accommodate these suggestions to help support our statements.

Reviewer #2 (Remarks to the Author):

The authors have made an effort to improve their manuscript and have been successful, within limits. I have an issue with broadscale, sometimes trivial statements on patterns seen in the omic data which do not necessarily deepen our knowledge but this is a more general point which authors may find hard to overcome. However, due to some bias with their own literature on macrophysiology in a latitudinal cline, their discussion of metabolism still ignores the rich literature on hypoxia adaptation in invertebrates, the detailed knowledge about differential mitochondrial densities and metabolic mechanisms and regulation in animal tissues in a latitudinal cline, the use of anaerobic metabolism including acetate in invertebrates at thermal limits etc. This should still be alleviated. Otherwise I am happy with the revisions and think the paper makes a valuable contribution to understanding the diversity of responses to ocean acidification in animal populations, identifying the position of the population within such range as one of the factors shaping that response diversity.

RE: We thank the reviewer for recognising our efforts in addressing the points raised. We have taken up their suggestion to include more detail from ‘the rich literature on hypoxia adaptation in invertebrates, the detailed knowledge about differential mitochondrial densities and metabolic mechanisms and regulation in animal tissues in a latitudinal cline, the use of anaerobic metabolism including acetate in invertebrates at thermal limits etc’ to support our discussion of metabolic responses. This includes the addition of the following new references: Morley, S A., Lurman, G. J., Skepper, J. N., Pörtner, H.-O. & Peck, L. S. Thermal plasticity of mitochondria: a latitudinal comparison between Southern Ocean molluscs. *Comp. Biochem. Phys. A.* 152, 423-430 (2009); Lucassen, M., Koschnick, N., Eckerle, L. G. & Pörtner, H.-O. Mitochondrial mechanisms of cold adaptation in cod (*Gadus morhua* L.) populations from different climatic zones. *J. Exp. Biol.* 209, 2462-2471 (2006); Johnston, I. A. Cold adaptation in marine organisms. *Philos. T. Roy. Soc. B.* 326, 655-667 (1990). **Reviewer #3 (Remarks to the Author):**

The authors have responded comprehensively on all queries and criticisms raised by the three reviewers. I have nothing further to add.

RE: We are thankful for this positive remark.

Two comments on the figures for the authors (and the editor) to consider:

Figure 1 is, in my opinion disconnected across panels with the current layout defying the purpose of grouping these panels. While I understand the authors' reluctance to display latitude on the y-axes of figure 1 panels b-d as this is not the convention to infer mechanism I think there is still a case for this to maximise the visual impact of the figure. I am sure the readers will be smart enough to understand if a succinct explanation/justification is added to the caption or methods. Similarly, while I understand that scale bars normally go from low to high, I think it would be more logical in Fig1a if the scale bar followed the latitudinal gradient from warm to cool.

RE: we have separated Figure 1 into Figure 1 and Figure 2. Figure 1 (old Fig. 1a) reports the map with the phylo-geographical information. Figure 2 (a, b and c) reports the physiology information panels (old Fig. 1b, c, d). We feel that this new presentation make things clearer and avoids any potential confusion from having latitude presented on both y and x axes in a better way than switching axes for the old Fig. 1.

Figure 2 has a lot of wasted blank space around all the panels and could be condensed somewhat.

RE: the figure has been remodelled and divided into two separate figures, whilst white spaces where reduced and panel quality improved.